# Why neural networks find simple solutions:
# the many regularizers of geometric complexity

**Benoit Dherin**[*]
Google
dherin@google.com

**Michael Munn**[*]
Google
munn@google.com

**Mihaela Rosca**
DeepMind, London
University College London
mihaelacr@deepmind.com

**David G.T. Barrett**
DeepMind, London
barrettdavid@deepmind.com

## Abstract

In many contexts, simpler models are preferable to more complex models and the control of this model complexity is the goal for many methods in machine learning such as regularization, hyperparameter tuning and architecture design. In deep learning, it has been difficult to understand the underlying mechanisms of complexity control, since many traditional measures are not naturally suitable for deep neural networks. Here we develop the notion of geometric complexity, which is a measure of the variability of the model function, computed using a discrete Dirichlet energy. Using a combination of theoretical arguments and empirical results, we show that many common training heuristics such as parameter norm regularization, spectral norm regularization, flatness regularization, implicit gradient regularization, noise regularization and the choice of parameter initialization all act to control geometric complexity, providing a unifying framework in which to characterize the behavior of deep learning models.

## 1 Introduction

Regularization is an essential ingredient in the deep learning recipe and understanding its impact on the properties of the learned solution is a very active area of research [36, 46, 79, 90]. Regularization can assume a multitude of forms, either added explicitly as a penalty term in a loss function [36] or implicitly through our choice of hyperparameters [7, 82, 85, 87, 89], model architecture [61, 63, 64] or initialization [34, 40, 41, 62, 74, 99, 100]. These forms are generally not designed to be analytically tractable, but in practice, regularization is often invoked in the control of model complexity, putting a pressure on a model to discover simple solutions more so than complex solutions.

To understand regularization in deep learning, we need to precisely define model 'complexity' for deep neural networks. Complexity theory provides many techniques for measuring the complexity of a model, such as a simple parameter count, or a parameter norm measurement [4, 25, 67, 76] but many of these measures can be problematic for neural networks [48, 97, 77]. The recently observed phenomena of 'double-descent' [12, 13, 75] illustrates this clearly: neural networks with high model complexity, as measured by a parameter count, can fit training data closely (sometimes interpolating the data exactly), while simultaneously having low test error [75, 97]. Classically, we expect that interpolation of training data is evidence of overfitting, but yet, neural networks seem to be capable of interpolation while also having low test error. It is often suggested that some form of implicit

---

[*]equal contribution

36th Conference on Neural Information Processing Systems (NeurIPS 2022).

regularization or explicit regularization is responsible for this, but how should we account for this in theory, and what complexity measure is most appropriate?

In this work, we develop a measure of model complexity, called Geometric Complexity (GC), that has properties that are suitable for the analysis of deep neural networks. We use theoretical and empirical techniques to demonstrate that many different forms of regularization and other training heuristics can act to control geometric complexity through different mechanisms. We argue that the geometric complexity provides a convenient proxy for neural network performance.

Our primary contributions are:

- We develop a computationally tractable notion of complexity (Section 2), which we call *Geometric Complexity* (GC), that has many close relationships with many areas in deep learning and mathematics including harmonic function theory, Lipschitz smoothness (Section 2), and regularization theory (Section 4).

- We provide evidence that common training heuristics keep the geometric complexity low, including: (i) common initialization schemes (Section 3) (ii) the use of overparametrized models with a large number of layers (Fig. 2) (iii) large learning rates, small batch sizes, and implicit gradient regularization (Section 5 and Fig. 4) (iv) explicit parameter norm regularization, spectral norm regularization, flatness regularization, and label noise regularization (Section 4 and Fig. 3)

- We show that the geometric complexity captures the double-descent behaviour observed in the test loss as model parameter count increases (Section 6 and Fig. 5).

The aim of this paper is to introduce geometric complexity, explore its properties and highlight its connections with existing implicit and explicit regularizers. To disentangle the effects studied here from optimization choices, we use stochastic gradient descent without momentum to train all models. We also study the impact of a given training heuristic on geometric complexity in isolation of other techniques to avoid masking effects. For this reason we do not use data augmentation or learning rate schedules in the main part of the paper. In the Supplementary Material (SM) we redo most experiments using SGD with momentum (Section C.7) and Adam (Section C.8) with very similar conclusion. We also observe the same behavior of the geometric complexity in a setting using learning rate schedule, data augmentation, and explicit regularization in conjunction to improve model performance (Section C.6). The exact details of all experiments in the main paper are listed in SM Section B. All additional experiment results and details can be found in SM Section C.

## 2    Geometric complexity and Dirichlet energy

Although many different forms of complexity measures have been proposed and investigated (e.g., [24, 48, 76]), it is not altogether clear what properties they should have, especially for deep learning. For instance, a number of them like the Rademacher complexity [54], the VC dimension [93], or the simple model parameter count focus on measuring the entire hypothesis space, rather than a specific function, which can be problematic in deep learning [48, 97]. Other measures like the number of linear pieces for ReLU networks [3, 86] or various versions of the weight matrix norms [76] measure the complexity of the model function independently from the task at hand, which is not desirable [81]. An alternative approach is to learn a complexity measure directly from data [60] or to take the whole training procedure over a dataset into account [75]. Recently, other measures focusing on the model function complexity over a dataset have been proposed in [30] and [67] to help explain the surprising generalization power of deep neural networks. Following that last approach and motivated by frameworks well established in the field of geometric analysis [50], we propose a definition of complexity related to the theory of harmonic functions and minimal surfaces. Our definition has the advantage of being computationally tractable and implicitly regularized by many training heuristics in the case of neural networks. It focuses on measuring the complexity of individual functions rather than that of the whole function space, which makes it different from the Radamacher or VC complexity.

**Definition 2.1.** *Let $g_\theta : \mathbb{R}^d \to \mathbb{R}^k$ be a neural network parameterized by $\theta$. We can write $g_\theta(x) = a(f_\theta(x))$ where $a$ denotes the last layer activation, and $f_\theta$ its logit network. The GC of the network*

*over a dataset $D$ is defined to be the discrete Dirichlet energy of its logit network:*

$$\langle f_\theta, D \rangle_G = \frac{1}{|D|} \sum_{x \in D} \|\nabla_x f_\theta(x)\|_F^2, \tag{1}$$

*where $\|\nabla_x f_\theta(x)\|_F$ is the Frobenius norm of the network Jacobian.*

Note that this definition is well-defined for any differentiable model, not only a neural network, and incorporates both the model function and the dataset over which the task is determined.

Next, we discuss how GC relates to familiar concepts in deep learning.

**Geometric complexity and linear models:** Consider a linear transformation $f(x) = Ax + b$ from $\mathbb{R}^d$ to $\mathbb{R}^k$ and a dataset $D = \{x_i\}_{i=1}^N$ where $x_i \in \mathbb{R}^d$. At each point $x \in D$, we have that $\|\nabla_x f(x)\|_F^2 = \|A\|_F^2$, hence the GC for a linear transformation is $\langle f_\theta, D \rangle_G = \|A\|_F^2$. Note, this implies that GC for linear transformations (and more generally, affine maps), is independent of the dataset $D$ and zero for constant functions. Furthermore, note that the GC in this setting coincides precisely with the L2 norm of the model weight matrix. Thus, enforcing an L2 norm penalty is equivalent to regularizing the GC for linear models (see Section 4 for more on that point).

**Geometric complexity and ReLU networks:** For a ReLU network $g_\theta : \mathbb{R}^d \to \mathbb{R}^k$ as defined in Definition 2.1, the GC over a dataset $D$ has a very intuitive form. Since a ReLU network parameterizes piece-wise linear functions [3], the domain can be broken into a partition of subsets $X_i \subset \mathbb{R}^d$ where $f_\theta$ is an affine map $A_i x + b_i$. Now denote by $D_i$ the points in the dataset $D$ that fall in the linear piece defined on $X_i$. For every point $x$ in $X_i$, we have that $\|\nabla_x f_\theta(x)\|_F^2 = \|A_i\|_F^2$. Since the $D_i$'s partition the dataset $D$, we obtain

$$\langle f_\theta, D \rangle_G = \sum_i \left( \frac{n_i}{|D|} \right) \|A_i\|_F^2, \tag{2}$$

where $n_i$ is the number of points in the dataset $D$ falling in $X_i$. We see from Eqn. (2) that for ReLU networks the GC over the whole dataset coincides exactly with the GC on a batch $B \subset D$, provided that the proportion of points in the batch falling into each of the the linear pieces are preserved. This makes the evaluation of the GC on large enough batches a very good proxy to the overall GC over the dataset, and computationally tractable during training.

**Geometric complexity and Lipschitz smoothness:** One way to measure the smoothness of a function $f : \mathbb{R}^d \to \mathbb{R}^k$ on a subset $X \subset \mathbb{R}^d$ is by its Lipschitz constant; i.e., the smallest $f_L \geq 0$ such that $\|f(x_1) - f(x_2)\| \leq f_L \|x_1 - x_2\|$, for all $x_1, x_2 \in X$. Intuitively, the constant $f_L$ measures the maximal amount of variation allowed by $f$ when the inputs change by a given amount. Using the Lipschitz constant, one can define a complexity measure of a function $f$ as the Lipschitz constant $f_L$ of the function over the input domain $\mathbb{R}^d$. Since $\|\nabla_x f(x)\|_F^2 \leq \min(k, d) \|\nabla_x f(x)\|_{op}^2 \leq \min(k, d) f_L^2$ where $\| \cdot \|_{op}$ is the operator norm, we obtain a general bound on the GC by the Lipschitz complexity:

$$\langle f, D \rangle_G = \frac{1}{|D|} \sum_{x \in D} \|\nabla_x f(x)\|_F^2 \leq \min(k, d) f_L^2. \tag{3}$$

While the Lipschitz smoothness of a function provides an upper bound on GC, there are a few fundamental differences between the two quantities. Firstly, GC is data dependent: a model can have low GC while having high Lipschitz constant due to the model not being smooth in parts of the space where there is no training data. Secondly, the GC can be computed exactly given a model and dataset, while for neural networks only loose upper bounds are available to estimate the Lipschitz constant.

**Geometric complexity, arc length and harmonic maps:** Let us start with a motivating example: Consider a dataset consisting of 10 points lying on a parabola in the plane and a large ReLU deep neural network trained via gradient descent to learn a function $f : \mathbb{R} \to \mathbb{R}$ that fits this dataset (Fig. 1). Throughout training the model function seems to attain minimal arc length for a given level of training error. Recalling the formula for arc length of $f$, which is the integral of over $[-1, 1]$ of $\sqrt{1 + f'(x)^2}$, and using the Taylor approximation $\sqrt{1 + x^2} \approx 1 + \frac{x^2}{2}$, it follows that minimizing the arc length is equivalent to minimizing the classic Dirichlet energy:

$$E(f) = \frac{1}{2} \int_\Omega \|\nabla_x f(x)\|^2 dx. \tag{4}$$

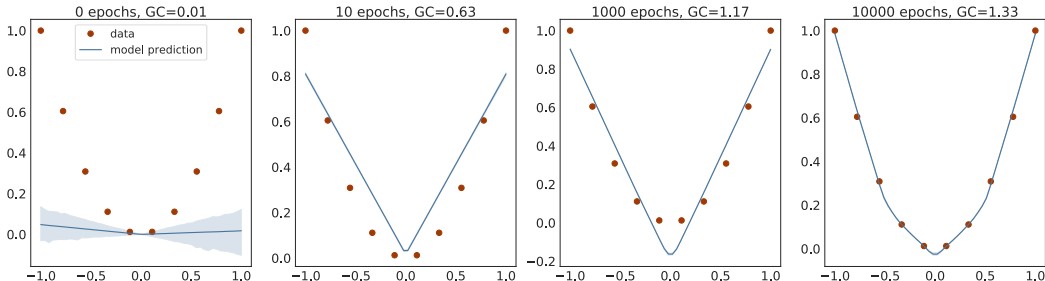

Figure 1: For a large MLP fitting 10 points, the complexity of the function being learned gradually grows in training, while avoiding unnecessary complexity by keeping the function arc length minimal.

where $\Omega = [-1, 1]$. The Dirichlet energy can be thought intuitively of as a measure of the variability of the function $f$ on $\Omega$. Its minimizers, subject to a boundary condition $f_{|\partial\Omega} = h$, are called *harmonic maps*, which are maps causing the "least intrinsic stretching" of the domain $\Omega$ [91]. The geometric complexity in Definition 2.1 is an unbiased estimator of a very related quantity

$$\mathbb{E}_X(\|\nabla_x f_\theta(x)\|_F^2) = \int_{\mathbb{R}^d} \|\nabla_x f_\theta(x)\|_F^2 \, p_X(x) dx, \tag{5}$$

where the domain $\Omega$ is replaced by the probability distribution of the features $p_X$ and the boundary condition is replaced by the dataset $D$. We could call the quantity defined in Eqn. (5) the *theoretical geometric complexity* as opposed to the *empirical geometric complexity* in Definition 2.1. The theoretical geometric complexity is very close to a complexity measure investigated in [78], where the Jacobian of the full network is considered rather than just the logit network as we do here. In their work, the expectation is also evaluated on the test distribution. They observe a correlation between this complexity measure and generalization empirically in a set of extensive experiments. In our work, we use the logit network (rather than the full network) and we evaluate the empirical geometric complexity on the train set (rather than on the test set) in order to derive theoretically that the implicit gradient regularization mechanism from [7] creates a regularizing pressure on GC (see Section 5).

In the remaining sections we provide evidence that common training heuristics do indeed keep the GC low, encouraging neural networks to find intrinsically simple solutions.

## 3 Impact of initialization on geometric complexity

Parameter initialization choice is an important factor in deep learning. Although we are free to specify exact parameter initialization values, in practice, a small number of default initialization schemes have emerged to work well across a wide range of tasks [34, 41]. Here, we explore the relationship between some of these initialization schemes and GC.

To begin, consider the one dimensional regression example that we introduced in Figure 1. In this experiment, we employed a standard initialization scheme to initialise the parameters: we sample them from a truncated normal distribution with variance inversely proportional to the number of input units and the bias terms were set to zero. We observe that the initialised function on the interval $[-1, 1]$ is very close to the zero function (Fig. 1), and the zero function has zero GC. This observation suggests that initialization schemes that have low initial GC are useful for deep learning.

To explore this further, we consider deep ReLU networks with larger input and output spaces initialized using the same scheme as above, and measure the GC of the resulting model. Specifically, consider the initialized ReLU network given by $f_{\theta_0} : \mathbb{R}^d \to \mathbb{R}^k$ with $d = 150528$ and $k = 1000$, with parameter initialization $\theta_0$, and varying network depth. We measure the ReLU network output size by recording the mean and maximum output values, evaluated using input values along the line $P_1 + (P_2 - P_1)x$ with $x \in [0, 1]$ between two diagonal points $P_1$ and $P_2$ of the hyper-cube $[-1, 1]^d$. We observe that these ReLU networks initialize to functions close to the zero function, and become progressively closer to a zero valued function as the number of layer increases (Fig. 2). For ReLU networks, this is not entirely surprising. With biases initialised to zero, we can express a ReLU network in a small neighborhood of a given point $x \in \mathbb{R}^d$ as a product of matrices $f_{\theta_0}(x) = W_1 P_2 W_2 P_3 W_3 \cdots P_l W_l x$, where the $W_i$'s are the weight matrices and the $P_i$'s are diagonal matrices

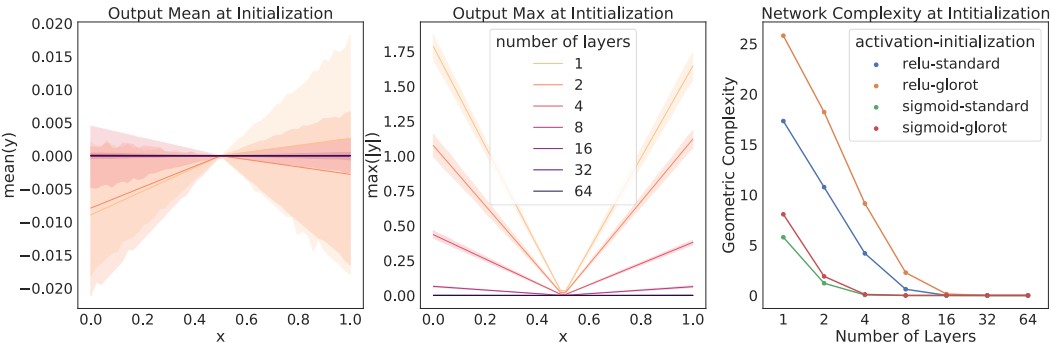

Figure 2: MLP's given by $y = f_{\theta_0}(x)$ initialize closer to the zero function and closer to zero GC, as the number of layers increases. **Left and Middle:** The ReLU MLP's initialized with the standard scheme are evaluated using input values along the line $P_1 + (P_2 - P_1)x$ with $x \in [0,1]$ between two diagonal points $P_1$ and $P_2$ of the hyper-cube $[-1,1]^d$. **Right:** GC is computed on a dataset $D$ of 100 normalized data points. All MLP's have 500 neurons per layer.

with 0 and 1 on their diagonals. This representation makes it clear that at initialization the ReLU network passes through the origin; i.e., $f_{\theta_0}(0) = 0$. Furthermore, with weight matrices initialised around zero, using a scaling that can reduce the spread of the distribution as the matrices grow, we can expect that deeper ReLU networks generated by multiplying a large number of small-valued weights, can produce output values close to zero (for input values taken from a hyper-cube $[-1,1]^d$). We extend these results further to include additional initialization schemes and experimentally confirm that the GC can be brought close to zero with a sufficient number of layers. In fact, this is true not only for ReLU networks with the standard initialization scheme, but for a number of other common activation functions and initialization setups [34, 41], and even on domains much larger than the normalized hyper-cube (Fig. 2 and SM Section C.1). Theoretically, it has been shown very recently in [5] (their Theorem 5), that under certain technical conditions, a neural network at random initialization will converge to a constant function (which has GC equal to zero) as the number of layers increases.

## 4 Impact of explicit regularization on geometric complexity

Next, we explore the relationship between GC and various forms of explicit regularization using a combination of theoretical and empirical results.

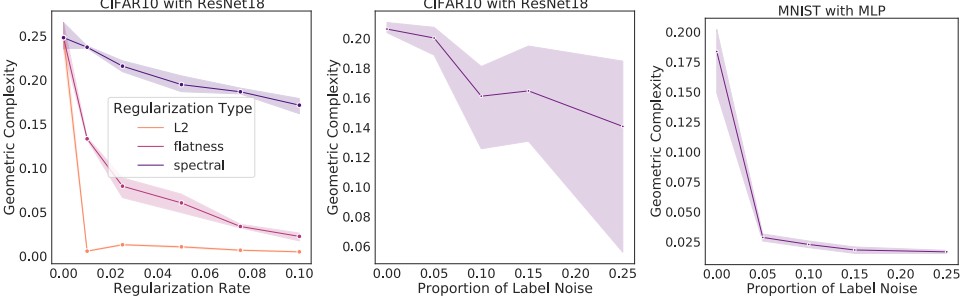

Figure 3: Explicit regularization and GC. **Left:** As the L2, flatness, and spectral regularization increase, GC decreases. (See SM Section C.5 for additional experiments on explicit regularization with ranges targeted to each regularization type.) **Middle and Right:** As label noise regularization increases, GC decreases.

**L2 regularization:** In L2 regularization, the Euclidean norm of the parameter vector is explicitly added to a loss function, so as to penalize solutions that have excessively large parameter values. This is one of the simplest, and most widely used forms of regularization. For linear models

$(f_\theta(x) = Ax + b)$, we saw in Section 2 that the GC coincides with the Frobenius norm of the matrix $A$. This means that standard L2 norm penalties on the weight matrix coincide in this case with a direct explicit regularization of the GC of the linear model. For non-linear deep neural networks, we cannot directly identify the L2 norm penalty with the model GC. However, in the case of ReLU networks, for each input point $x$, the network output $y$ can be written in a neighborhood of $x$ as $y = P_l W_l \ldots P_1 W_1 x + c$, where $c$ is a constant, the $P_i$'s are diagonal matrices with 0 and 1 on the diagonal, and the $W_i$'s are the network weight matrices. This means that the derivative at $x$ of the network coincides with the matrix $P_l W_l \ldots P_1 W_1$. Therefore, the GC is just the Frobenius norm of the product of matrices. Now, an L2 penalty $\|W_l\|_F^2 + \cdots + \|W_1\|_F^2$ encourages small numbers in the values of the weight matrices, which in turn is likely to encourage small numbers for the values in the product $P_l W_l \ldots P_1 W_1$, resulting in a lower GC.

We can demonstrate this relationship empirically, by training a selection of neural networks with L2 regularization, each with a different regularization strength. We measure the GC for each network at the time of maximum test accuracy. We observe empirically that strengthening L2 regularization coincides with a decrease in GC values (Fig. 3).

**Lipschitz regularization via spectral norm regularization:**    A number of explicit regularization schemes have been used to tame the Lipschitz smoothness of the model function and produce smoother models with reduced test error [2, 3, 26, 28, 39, 53, 71, 81, 96]. Smoothness regularization has also been shown to be beneficial outside the supervised learning context, in GANs [17, 71, 98] and reinforcement learning [14, 35]. One successful approach to regularising the Lipschitz constant of a neural network with 1-Lipschitz activation functions (e.g. ReLU, ELU) is to constrain the spectral norm of each layer of the network (i.e., the maximal singular values $\sigma_{\max}(W_i)$ of the weight matrices $W_i$), since the product of the spectral norms of the networks weight matrices is an upper bound to the Lipschitz constant of the model: $f_L \leq \sigma_{\max}(W_1) \cdots \sigma_{\max}(W_l)$. Using inequality (3), we see that any approach that constrains the Lipschitz constant of the model constrains GC. To confirm this theoretical prediction, we train a ResNet18 model on CIFAR10 [58] and regularize using spectral regularization [96], an approach which adds the regularizer $\frac{\lambda}{2} \sum_l (\sigma_{\max}(W_l))^2$ to the model loss function. We observe that GC decreases as the strength of spectral regularization $\lambda$ increases (Fig. 3).

**Noise regularization:**    The addition of noise during training is known to be an effective form of regularization. For instance, it has been demonstrated [15] that the addition of noise to training labels during the optimization of a least-square loss using SGD exerts a regularising pressure on $\sum_{(x,y)\in D} \|\nabla_\theta f_\theta(x)\|^2 / |D|$. Here, we demonstrate empirically that the GC of a ResNet18 trained on CIFAR10 reduces as the proportion of label noise increases (Fig. 3, middle). The same is true of an MLP trained on MNIST [22] (Fig. 3, right). In Section 5, we provide a theoretical argument which justifies these experiments, showing that a regularizing pressure on $\|\nabla_\theta f_\theta(x)\|^2$ transfers to a regularizing pressure on $\|\nabla_x f_\theta(x)\|^2$. Thus, label noise in SGD in turn translates into a regularizing pressure on the GC in the case of neural networks.

**Flatness regularization:**    In flatness regularization, an explicit gradient penalty term, $\|\nabla_\theta L_B\|^2$ is added to the loss (where $L_B$ is the loss evaluated across a batch $B$). It has been observed in practice that flatness regularization can be effective in many deep learning settings, from supervised learning [31, 87] to GAN training [6, 70, 73, 80, 82]. Flatness regularization penalizes learning trajectories that follow steep slopes across the loss surface, thereby encouraging learning to follow shallower slopes toward flatter regions of the loss surface. We demonstrate empirically that GC decreases as the strength of flatness regularization increases (Fig. 3). In the next section, we will provide a theoretical argument that flatness regularization can control GC.

**Explicit GC regularization and Jacobian regularization:**    All the forms of regularization above have known benefits for improving the test accuracy in deep learning. As we saw, they all also implicitly regularize GC. This raises the question as whether regularizing for GC directly and independently of any other mechanism is sufficient to improve model performance. Namely, we can add GC computed on the batch to the loss $L_{\text{reg}}(\theta) = L_B(\theta) + \alpha/B \sum_{x \in B} \|\nabla_x f_\theta(x)\|_F^2$. This is actually a known form of explicit regularization, called **Jacobian regularization**, which has been correlated with increased generalization but also robustness to input shift [46, 90, 95, 96]. [90] specifically shows that adding Jacobian regularization to the loss function can lead to an increase in test set accuracy (their Tables III, IV, and V). In SM Section C.4 we train a MLP on MNIST and a

ResNet18 on CIFAR10 regularized explicitly with the geometric complexity. We observe an increase of test accuracy and a decrease of GC with more regularization. Related regularizers include *gradient penalties* of the form $\sum_x (\|\nabla_x f_\theta(x)\| - K)^2$ which have been used for GAN training [28, 39, 52]. We leave the full investigation of the importance of GC outside supervised learning for future work.

## 5 Impact of implicit regularization on geometric complexity

Implicit regularization is a hidden form of regularization that emerges as a bi-product of model training. Unlike explicit regularization, it is not explicitly added to a loss function. In deep learning settings where no explicit regularization is used, it is the only form of regularization. Here, we use a combination of theoretical and empirical results to argue that some recently identified implicit regularization mechanisms in gradient descent [7, 65, 87] exert a regularization pressure on geometric complexity. Our argument proceeds as follows: 1) we identify a mathematical term (the implicit gradient regularization term) that characterizes implicit regularization in gradient descent, 2) we demonstrate that this term depends on model gradients, 3) we identify the conditions where model gradient terms apply a regularization pressure on geometric complexity.

**Step 1:** The implicit regularization that we consider emerges as a bi-product of the discrete nature of gradient descent updates. In particular it has been shown that a discrete gradient update $\theta' = \theta - h\nabla_\theta L_B(\theta)$ over a batch of data implicitly minimizes a modified loss, $\widetilde{L}_B = L_B + \frac{h}{4}\|\nabla_\theta L_B\|^2$, where the second term is called the Implicit Gradient Regularizer (IGR) [7]. Gradient descent optimization is better characterized as a continuous flow along the gradient of the modified loss, rather than the original unmodified loss. By inspection, the IGR term implicitly regularizes training toward trajectories with smaller loss gradients toward flatter region on the loss surface [7] .

**Step 2:** Next, we develop this implicit regularizer term for a multi-class classification cross-entropy loss term. We can write (see SM Section A.2 for details):

$$\widetilde{L}_B = L_B + \frac{h}{4B}\left(\frac{1}{B}\sum_{x,i}\epsilon_x^i(\theta)^2\|\nabla_\theta f_\theta^i(x)\|^2\right) + \frac{h}{4}A_B(\theta) + \frac{h}{4}C_B(\theta), \tag{6}$$

where $C_B(\theta)$ measures the *batch gradient alignment*

$$C_B(\theta) = \frac{1}{B^2}\sum_{(x,y)\neq(x',y')}\langle\nabla_\theta L(x,y,\theta), \nabla_\theta L(x',y',\theta)\rangle. \tag{7}$$

and $A_B(\theta)$ measures the *label gradient alignment*:

$$A_B(\theta) = \frac{1}{B^2}\sum_{x\in B}\sum_{i\neq j}\langle\epsilon_x^i\nabla_\theta f_\theta^i(x), \epsilon_x^j\nabla_\theta f_\theta^j(x)\rangle. \tag{8}$$

where $\epsilon_x^i(x) = a(f_\theta(x))^i - y^i$ is the signed residual and $a$ denotes the activation function of the last layer. Note that this residual term arises in our calculation using $\nabla_\theta L(x,y,\theta) = \epsilon_x^1(\theta)\nabla_\theta f_\theta^1(x) + \cdots + \epsilon_x^k(\theta)\nabla_\theta f_\theta^k(x)$ (see SM Section A.1, Eqn. (14)). This development also extends to other widely used loss functions such as the cross entropy and least-square loss.

**Step 3:** Now, from the modified loss in Eqn. (6), we can observe the conditions under which SGD puts an implicit pressure on the gradient norms $\|\nabla_\theta f_\theta^i(x)\|^2$ to be small: the batch gradient alignment and label gradient alignment terms must be small relative to the gradient norms, or positive valued.

We also derive conditions under which this implicit pressure on $\|\nabla_\theta f_\theta(x)\|_F^2$ transfers to a regularizing pressure on $\|\nabla_x f_\theta(x)\|_F^2$ (see SM Section A.3 for the proof):.

**Theorem 5.1.** *Consider a logit network $f_\theta : \mathbb{R}^d \to \mathbb{R}^k$ with $l$ layers, then we have the following inequality*

$$\|\nabla_x f_\theta(x)\|_F^2 \leq \frac{\|\nabla_\theta f_\theta(x)\|_F^2}{T_1^2(x) + \cdots + T_l^2(x)}, \tag{9}$$

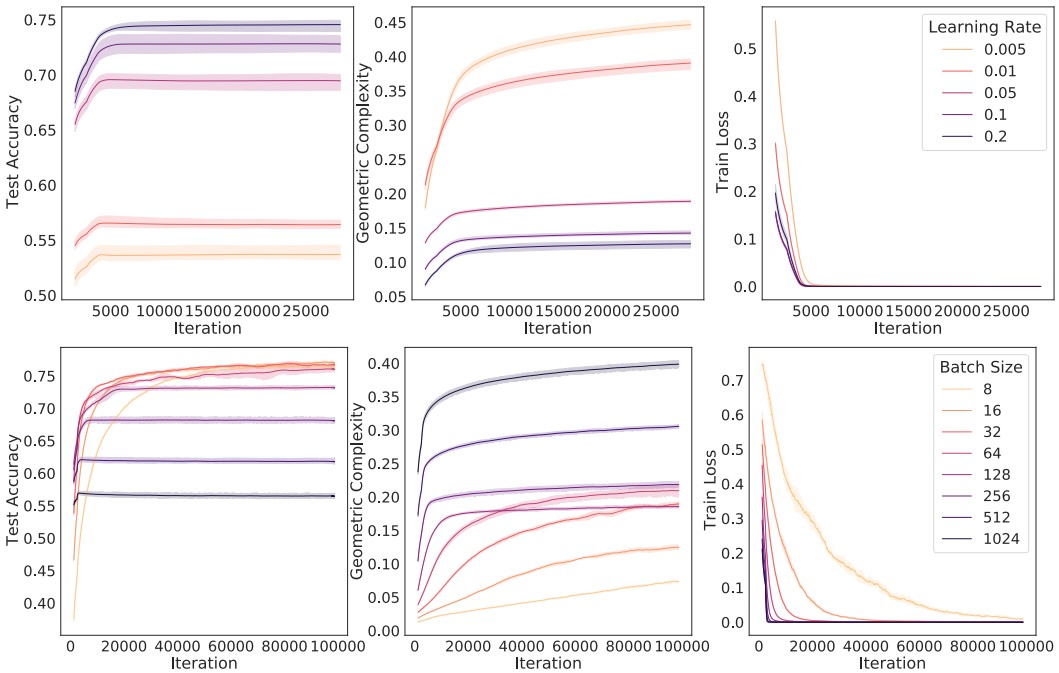

Figure 4: Impact of IGR when training ResNet18 on CIFAR10. **Top row:** As IGR increases through higher learning rates, GC decreases. **Bottom row:** Similarly, lower batch size leads to decreased GC.

*where $T_i$ is the transfer function for layer $i$ given by*

$$T_i(x,\theta) = \frac{1}{\sqrt{\min(d,k)}} \frac{\sqrt{1+\|h_i(x)\|_2^2}}{\sigma_{\max}(W_i)\sigma_{\max}(h_i'(x))},\tag{10}$$

*where $h_i$ is the subnetwork to layer $i$ and $\sigma_{\max}(A)$ is the maximal singular value of matrix $A$ (i.e., its spectral norm).*

Here, we can see that for any settings where the sum $T_1^2(x) + \cdots + T_l^2(x)$ of squared transfer functions diminishes slower than the gradient norm $\|\nabla_\theta f_\theta(x)\|_F^2$ during training, we expect that implicit gradient regularization will apply a regularization pressure on GC.

An immediate prediction arising under these conditions is that the size of the regularization pressure on geometric complexity will depend on the implicit regularization rate $h/B$ in Eqn. (6) (Note that the ratio $h/B$ has been linked to implicit regularization strength in many instances [37, 88, 69, 59, 65]). Specifically, under these conditions, networks trained with larger learning rates or smaller batch sizes, or both, will apply a stronger regularization pressure on geometric complexity,

We test this prediction by performing experiments on ResNet18 trained on CIFAR10 and show results in Figure 4. The results show that while all models achieve a zero train loss, consistently the higher the learning rate, the higher the test accuracy and the lower the GC. Similarly, we observe that the lower the batch size, the lower the GC and the higher the test accuracy.

The increased performance of lower batch sizes has been long studied in deep learning, under the name 'the generalisation gap' [45, 51]. Crucially however, this gap was recently bridged [32], showing that full batch training can achieve the same test set performance as mini-batch gradient descent. To obtain these results, they use an explicit regularizer similar to the implicit regularizer in Eqn. (6), introduced to compensate for the diminished implicit regularization in the full batch case. Their results further strengthen our hypothesis that implicit regularization via GC results in improved generalisation.

# 6 Geometric complexity and double descent

When complexity is measured using a simple parameter count, a double descent phenomena has been consistently observed: as the number of parameters increases, the test accuracy decreases at first, before increasing again, followed by a second descent toward a low test error value [12, 13, 75]. An excellent overview of the double descent phenomena in deep learning can be found in [12], together with connections to smoothness as an inductive bias of deep networks and the role of optimization.

To explore the double descent phenomena using GC we follow the set up introduced in [75]: we train multiple ResNet18 networks on CIFAR10 with increasing layer width, and show results in Fig. 5. We make two observations: first, like the test loss, GC follows a double descent curve as width increases; second, when plotting GC against the test loss, we observe a U-shape curve, recovering the traditional expected behaviour of complexity measures. Importantly, we observe the connection between the generalisation properties of overparametrized models and GC: after the critical region, increase in model size leads to a decrease in GC. This provides further evidence that GC is able to capture model capacity in a meaningful way, suggestive of a reconciliation of traditional complexity theory and deep learning.

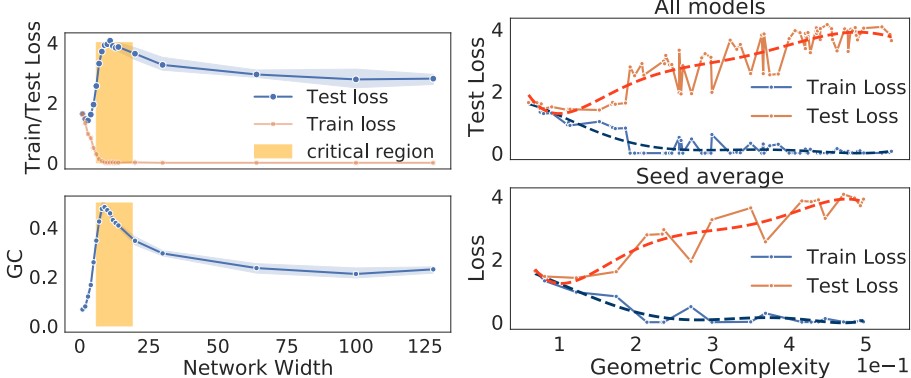

Figure 5: Double descent and GC. **Left:** GC captures the double descent phenomenon. **Right:** GC captures the traditional U-shape curve, albeit with some noise, showing (top) GC vs Test Loss for all models and (bottom) GC vs Test Loss averaged across different seeds. We fit a 6 degree polynomial to the curves to showcase the trend.

# 7 Related and Future Work

The aim of this work is to introduce a measure which captures the complexity of neural networks. There are many other approaches aiming at doing so, ranging from naive parameter count and more data driven approaches [47, 60, 67] to more traditional measures such as the VC dimension and Rademacher complexity [11, 48, 57, 92] which focus on the entire hypothesis space. [77] analyzes existing measures of complexity and shows that they increase as the size of the hidden units increases, and thus cannot explain behaviours observed in over-parameterized neural networks (their Figure 5). [48] performs an extensive study of existing complexity measures and their correlation with generalization. They find that flatness has the strongest positive connection with generalization. [9] provides a generalization bound depending on classification margins and the product of spectral norms of the model's weights and shows how empirically the product of spectral norms correlates with empirical risk. [20] connects Lipschitz smoothness with overparametrization, by providing a probabilistic lower bound on the model's Lipschitz constant based on its training performance and the inverse of its number of parameters. In concurrent work [29] discusses the connection between the Jacobian and the Hessian of the *loss* with respect to inputs, namely the empirical average of $\|\nabla_x L\|_2$ and $\|\nabla_x^2 L\|_2$ over the training set and shows that they follow a double-descent curve. [78] investigates empirically a complexity measure similar to GC using the Jacobian norm of the full network (rather than the logit network) and evaluating it on test distribution (rather than the train distribution). They show a correlation between this measure and generalization in an extensive set of experiments.

As we saw in Fig. 1, the interpolating ReLU network with minimal GC is also the piecewise linear function with minimal volume or length [23]. In 1D this minimal function can be described only with the information given by the data points. This description with minimal information is reminiscent of the Kolmogorov complexity [84] and the minimum description length [43], and we believe that the exact relationship between these notions, GC, and minimal volume is worth investigating. Similarly, [1] argues that flat solutions have low information content, and for neural networks these flat regions are also the regions of low loss gradient and thus of low GC, as explained by the Transfer Theorem in Section 5. Another recent measure of complexity is the *effective dimension* which is defined in relation to the training data, but is computed using the spectral decomposition of the loss Hessian [67], making flat regions in the loss surface also regions of low complexity w.r.t. this measure. This hints toward the effective dimension being related to GC. Note that similarly to GC, the effective dimension can also capture the double descent phenomena. The effective dimension is an efficient mechanism for model selection [67], which our experiments seem also to indicate may be the case for GC. Note that the generalized degrees of freedom (which considers the sensitivity of a classifier to the labels, rather than to the features as GC does) explored in [38] in the context of deep learning also captures the double-descent phenomena.

GC is close to considerations about smoothness [81] and the Sobolev norm implicit regularization [65]. While GC and Lipchitz smoothness are connected, here we focused on a tractable quantity for neural networks and its connections with existing regularizers. Smoothness regularization has been particularly successful in the GAN literature [18, 39, 52, 71, 98], and the connection between Lipschitz smoothness and GC begs the question of whether their success is due to their implicit regularization of GC, but we leave the application of GC outside supervised learning for future work.

The Dirichlet energy in Eqn. (4) is a well-known quantity in harmonic function theory [27] and a fundamental concept throughout mathematics, physics and, more recently, 3D modeling [91], manifold learning [19], and image processing [33, 83]. Minimizers of the Dirichlet energy are harmonic and thus enjoy certain guarantees in regularity; i.e., any such solution is a smooth function. It has been demonstrated that for mean squared error regression on wide-shallow ReLU networks, gradient descent is biased towards smooth functions at interpolation [49]. Similarly, our work suggests that neural networks, through a mechanism of implicit regularization of GC, are biased towards minimal Dirichlet energy and thus encourage smooth interpolation. It may be interesting to understand how the relationship between GC, Dirichlet energy, and harmonic theory can help futher improve the learning process, in particular in transfer learning, or in out-of-distribution generalization.

## 8   Discussion

In terms of *limitations*, the theoretical arguments presented in this work have a focus on ReLU activations and DNN architectures, with log-likelihood losses coming from the exponential family, such as multi-dimensional regression with least-square losses, or multi-class classification with cross-entropy loss. The experimental results are obtained on the image datasets MNIST and CIFAR using DNN and ResNet architectures.

In terms of *societal impact*, we are not introducing new training methods, but focus on providing a better understanding of the impact of common existing training methods. We hope that this understanding will ultimately lead to more efficient training techniques. While existing training methods may have their own risk, we do not foresee any potential negative societal impact of this work.

In conclusion, altogether, geometric complexity provides a useful lens for understanding deep learning and sheds light into why neural networks are able to achieve low test error with highly expressive models. We hope this work will encourage further research around this new connection, and help to better understand current best practices in model training as well as discover new ones.

## Acknowledgments and Disclosure of Funding

We would like to thank Chongli Qin, Samuel Smith, Soham De, Yan Wu, and the reviewers for helpful discussions and feedback as well as Patrick Cole, Xavi Gonzalvo, and Shakir Mohamed for their support.

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
