# Supplementary Material

## A  Proofs for Section 5

In Section A.1 we derive that for a large class of models comprising regression models with the least-square loss and classification models with the cross-entropy loss, the norm square of the loss gradients has a very particular form (Eqn. 17) that we need to develop the modified loss in Eqn. 6 in Step 2 of Section 5. The detail of this development is given in Section A.2. In Section A.3, we give the proof of Thm. 5.1 also needed in Step 2 of Section 5, which bounds the Frobenius norm of the network Jacobian w.r.t. the input with the network Jacobian w.r.t. the parameters.

### A.1  Gradient structure in the exponential family

In order to be able to treat both regression and classification on the same footing, we need to introduce the notion of a transformed target $y = \phi(z)$. In regression $\phi$ is typically the identity, while for classification $\phi$ maps the labels $z \in \{1, \ldots, k\}$ onto their one-hot-encoded version $\phi(z) \in \mathbb{R}^k$. We will also need to distinguish between the *logit neural network* $f_\theta : \mathbb{R}^d \to \mathbb{R}^k$ from the *response function* $g_\theta(x) = a(f_\theta(x))$ that models the transformed target. Typically, $a$ is the activation function of the last layer. For instance, for regression both the neural network and its response function

coincide, $a$ being the identity, while for classification $f_\theta(x)$ are the logits and $a$ is typically a sigmoid or a softmax function. Now both regression and classification losses at a data point $(x, y)$ can be obtained as the negative log-likelihood of a conditional probability model

$$L(x, y, \theta) = -\log P(y|x, \theta), \tag{11}$$

The conditional probability model for both regression and classification has the same structure. It is obtained by using the neural network to estimate the natural parameter vector $\eta = f_\theta(x)$ of an exponential family distribution:

$$P(y|x, \theta) = h(y) \exp\left(\langle y, f_\theta(x)\rangle - S(f_\theta(x))\right), \tag{12}$$

where $S(\eta)$ is the log-partition function. For models in the exponential family, the distribution mean coincides with the gradient of the log-partition function: $E(Y|\eta) = \nabla_\eta S(\eta)$. The response function (which is the mean of the conditional distribution) is of the form $g_\theta(x) = \nabla_\eta S(f_\theta(x))$, and the last layer is thus given by $a(\eta) = \nabla_\eta S(\eta)$. For these models, the log-likelihood loss at a data point has the simple form

$$L(x, y, \theta) = S(f_\theta(x)) - \langle y, f_\theta(x)\rangle + \text{constant}. \tag{13}$$

We then obtain immediately that the loss derivative w.r.t. to the parameters and w.r.t. to the input can be written as a sum of the the corresponding network derivatives weighted by the signed residuals:

$$\nabla_\theta L(x, y, \theta) = \epsilon_x^1(\theta)\nabla_\theta f_\theta^1(x) + \cdots + \epsilon_x^k(\theta)\nabla_\theta f_\theta^k(x), \tag{14}$$

$$\nabla_x L(x, y, \theta) = \epsilon_x^1(\theta)\nabla_x f_\theta^1(x) + \cdots + \epsilon_x^k(\theta)\nabla_x f_\theta^k(x), \tag{15}$$

where the $\epsilon^i$'s are the signed residual, that is, the $i^{th}$ components of the signed error between the response function and the transformed target:

$$\epsilon_x^i(\theta) = a^i(f_\theta(x)) - y^i. \tag{16}$$

This means that the square norm of these gradients can be written as

$$\|\nabla_\theta L(x, y, \theta)\|^2 = \sum_i \epsilon_x^i(\theta)^2 \|\nabla_\theta f_\theta^i(x)\|^2 + A_\theta(x, y, \theta) \tag{17}$$

$$\|\nabla_x L(x, y, \theta)\|^2 = \sum_i \epsilon_x^i(\theta)^2 \|\nabla_x f_\theta^i(x)\|^2 + A_x(x, y, \theta), \tag{18}$$

where $A_\theta$ and $A_x$ are the gradient alignment terms:

$$A_\theta(x, y, \theta) = \sum_{i \neq j} \langle \epsilon_x^i \nabla_\theta f_\theta^i(x), \epsilon_x^j \nabla_\theta f_\theta^j(x)\rangle \tag{19}$$

$$A_x(x, y, \theta) = \sum_{i \neq j} \langle \epsilon_x^i \nabla_x f_\theta^i(x), \epsilon_x^j \nabla_x f_\theta^j(x)\rangle. \tag{20}$$

### A.1.1 Multi-class classification

Consider the multinoulli distribution where a random variable $Z$ can take values in $k$ classes, say $z \in \{1, \ldots, k\}$ with probabilities $p_1, \ldots, p_k$ for each class respectively. Let the transformed target map $y = \phi(z)$ associate to a class $i$ its one-hot-encoded vector $y$ with the $i^{th}$ component equal to 1 and all other components zero. The multinoulli density can then be written as $P(y) = p_1^{y_1} \cdots p_k^{y_k}$ which can be re-parameterized, showing that the multinoulli is a member of the exponential family distribution:

$$
\begin{aligned}
P(y) &= \exp(\log(p_1^{y_1} \cdots p_l^{y_l})) \\
&= \exp\left(y_1 \log p_1 + \cdots + y_k \log p_k\right) \\
&= \exp\left(y_1 \log p_1 + \cdots + (1 - \sum_{l=1}^{k-1} y_l) \log p_k\right) \\
&= \exp\left(y_1 \log \frac{p_1}{p_k} + \cdots + y_{k-1} \log \frac{p_{k-1}}{p_k} + \cdots + \log p_k\right)
\end{aligned}
$$

We can now express the natural parameter vector $\eta$ in terms of the class probabilities:

$$\eta_i := \log \frac{p_i}{p_k} \quad \text{for } i = 1, 2, \ldots, k. \tag{21}$$

Note, $\eta_k = 0$. Taking the exponential of that last equation, and summing up, we obtain that $1/p_k = \sum_{i=1}^{k} e^{\eta_i}$, which we use to express the class probabilities in terms of the canonical parameters:

$$p_i = \frac{e^{\eta_i}}{\sum_l e^{\eta_l}}. \tag{22}$$

Since $1/p_k = \sum_{i=1}^{k} e^{\eta_i}$, we obtain that the log-partition function is

$$S(\eta) = \log \sum_{i=1}^{k} e^{\eta_i}, \tag{23}$$

whose derivative is the softmax function:

$$a(\eta) = \nabla_\eta S(\eta) = \frac{e^{\eta}}{\sum_{i=1}^{k} e^{\eta_i}}. \tag{24}$$

Now, if we estimate the natural parameter $\eta$ with a neural network $\eta = f_\theta(x)$, we obtain the response

$$E(Y \mid x, \theta) = \nabla_\eta S(f_\theta(x)) = a(f_\theta(x)), \tag{25}$$

## A.2 Computing the expanded modified loss from Section 5

In this section we verify that expanding the modified loss in Step 1. of Section 5, i.e., $\widetilde{L}_B = L_B + \frac{h}{4}\|\nabla L_B\|^2$, yields Eqn.(6). Namely, we show that

$$\widetilde{L}_B = L_B + \frac{h}{4B}\left(\frac{1}{B}\sum_{x,i}\epsilon_x^i(\theta)^2\|\nabla_\theta f_\theta^i(x)\|^2\right) + \frac{h}{4}A_B(\theta) + \frac{h}{4}C_B(\theta), \tag{26}$$

where

$$A_B(\theta) = \frac{1}{B^2}\sum_{x \in B}\sum_{i \neq j}\langle \epsilon_x^i \nabla_\theta f_\theta^i(x), \epsilon_x^j \nabla_\theta f_\theta^j(x)\rangle \tag{27}$$

$$C_B(\theta) = \frac{1}{B^2}\sum_{(x,y) \neq (x',y')}\langle \nabla_\theta L(x, y, \theta), \nabla_\theta L(x', y', \theta)\rangle. \tag{28}$$

It suffices to show

$$\|\nabla_\theta L_B(\theta)\|^2 = \frac{1}{B^2}\sum_{x,i}\epsilon_x^i(\theta)^2\|\nabla_\theta f_\theta^i(x)\|^2 \tag{29}$$

$$+ \frac{1}{B^2}\sum_{x \in B}\sum_{i \neq j}\langle \epsilon_x^i \nabla_\theta f_\theta^i(x), \epsilon_x^j \nabla_\theta f_\theta^j(x)\rangle \tag{30}$$

$$+ \frac{1}{B^2}\sum_{(x,y) \neq (x',y')}\langle \nabla_\theta L(x, y, \theta), \nabla_\theta L(x', y', \theta)\rangle. \tag{31}$$

Indeed this follows directly by computation. Firstly, note that

$$\|\nabla_\theta L_B(\theta)\|^2 = \left\langle \nabla_\theta \frac{1}{B} \sum_{x \in B} L(x, y, \theta), \nabla_\theta \frac{1}{B} \sum_{x' \in B} L(x', y', \theta) \right\rangle \tag{32}$$

$$= \frac{1}{B^2} \sum_{x \in B} \langle \nabla_\theta L(x, y, \theta), \nabla_\theta L(x, y, \theta) \rangle \tag{33}$$

$$+ \frac{1}{B^2} \sum_{(x,y) \neq (x',y')} \langle \nabla_\theta L(x, y, \theta), \nabla_\theta L(x', y', \theta) \rangle \tag{34}$$

$$= \frac{1}{B^2} \sum_{x \in B} \|\nabla_\theta L(x, y, \theta)\|^2 \tag{35}$$

$$+ \frac{1}{B^2} \sum_{(x,y) \neq (x',y')} \langle \nabla_\theta L(x, y, \theta), \nabla_\theta L(x', y', \theta) \rangle. \tag{36}$$

Now, by (17) and (19), the last equality can be simplified so that

$$\|\nabla_\theta L_B(\theta)\|^2 = \frac{1}{B^2} \sum_{x \in B} \sum_i \epsilon_x^i(\theta)^2 \|\nabla_\theta f_\theta^i(x)\|^2 \tag{37}$$

$$+ \frac{1}{B^2} \sum_{x \in B} \sum_{i \neq j} \langle \epsilon_x^i \nabla_\theta f_\theta^i(x), \epsilon_x^j \nabla_\theta f_\theta^j(x) \rangle \tag{38}$$

$$+ \frac{1}{B^2} \sum_{(x,y) \neq (x',y')} \langle \nabla_\theta L(x, y, \theta), \nabla_\theta L(x', y', \theta) \rangle. \tag{39}$$

This completes the computation.

### A.3 The Transfer Theorem

To frame the statement and proof of the Transfer Theorem, let's begin by setting up and defining some notation. Consider a deep neural network $f_\theta : \mathbb{R}^d \to \mathbb{R}^k$ parameterized by $\theta$ and consisting of $l$ layers stacked consecutively. We can express $f_\theta$ as

$$f_\theta(x) = f_l \circ f_{l-1} \circ \cdots \circ f_1(x), \tag{40}$$

where $f_i(z) = a_i(w_i z + b_i)$ denotes the $i$-th layer of the network defined by the weight matrix $w_i$, the bias $b_i$ and the layer activation function $a_i$ which acts on the output $z$ of the previous layer. In this way, we can write $\theta = (w_1, b_1, w_2, b_2, \ldots, w_l, b_l)$ to represent all the learnable parameters of the network.

Next, let $h_i(x)$ denote the sub-network from the input $x \in \mathbb{R}^d$ up to and including the output of layer $i$; that is,

$$h_i(x) = a_i(w_i h_{i-1}(x) + b_i), \text{ for } i = 1, 2, \ldots, l \tag{41}$$

where we understand $h_0(x)$ to be $x$. Note that, by this notation, $h_l(x)$ represents the full network; i.e., $h_l(x) = f_\theta(x)$.

At times it will be convenient to consider $f_\theta(x)$ as dependent only on a particular layer's parameters, for example $w_i$ and $b_i$, and independent of all other parameter values in $\theta$. In this case, we will use the notation $f_{w_i}$ or $f_{b_i}$ to represent $f_\theta$ as dependent only on $w_i$ or $b_i$ (resp.). Using the notation above, for each weight matrix $w_1, w_2, \ldots, w_l$, we can rewrite $f_{w_i}(x)$ as (similarly, for $f_{b_i}(x)$)

$$f_{w_i}(x) = g_i(w_i h_{i-1}(x) + b_i), \text{ for } i = 1, 2, \ldots, l \tag{42}$$

where each $g_i(z)$ denotes the remainder of the full network function $f_\theta(x)$ following the $i$-th layer. That is to say, $g_i$ represents the part of the network deeper than the $i$-th layer (i.e., from layer $i$ to the output $f_\theta(x)$) and $h_i$ represents the part of the network shallower than the $i$-th layer (i.e., from the input $x$ up to layer $i$).

We are now ready to state our main Theorem:

**Theorem A.1** (Transfer Theorem). *Consider a network $f_\theta : \mathbb{R}^d \to \mathbb{R}^k$ with $l$ layers parameterized by $\theta = (w_1, b_1, \ldots, w_l, b_l)$, then we have the following inequality*

$$\|\nabla_x f_\theta(x)\|_F^2 \le \frac{\|\nabla_\theta f_\theta(x)\|_F^2}{T_1^2(x, \theta) + \cdots + T_l^2(x, \theta)}, \tag{43}$$

*where $T_i(x, \theta)$ is the transfer function for layer $i$ given by*

$$T_i(x, \theta) = \frac{1}{\sqrt{\min(d, k)}} \frac{\sqrt{1 + \|h_{i-1}(x)\|_2^2}}{\sigma_{\max}(w_i)\sigma_{\max}(\nabla_x h_{i-1}(x))}, \tag{44}$$

*where $h_i$ is the subnetwork to layer $i$ and $\sigma_{\max}(A)$ is the maximal singular value of matrix $A$ (i.e., its spectral norm).*

The proof of Theorem A.1, inspired by the perturbation argument in [85], follows from the two following Lemmas. The idea is to examine the layer-wise structure of the neural network to compare the gradients $\nabla_{w_i} f_\theta$ with the gradients $\nabla_x f_\theta$. Due to the nature of $f_\theta(x)$ and its dependence on the inputs $x$ and parameters $\theta$, we show that at each layer $i$, a small perturbation of the inputs $x$ of the model function $f_\theta(x)$ transfers to a small perturbation of the weights $w_i$.

**Lemma A.2.** *Let $f_\theta : \mathbb{R}^d \to \mathbb{R}^k$ represent a deep neural network consisting of $l$ consecutive dense layers. Using the notation above, for $i = 1, 2, \ldots, l$, we have*

$$\|\nabla_x f_\theta(x)\|_2^2 \left( \frac{\|h_{i-1}(x)\|_2}{\|w_i\|_2 \|\nabla_x h_{i-1}(x)\|_2} \right)^2 \le \|\nabla_{w_i} f_\theta(x)\|_2^2, \tag{45}$$

*where $\|\cdot\|_2$ denotes the $L^2$ operator norm when applied to matrices and the $L^2$ norm when applied to vectors.*

*Proof.* For $i = 1, 2, \ldots, l$ and following the notation above, the model function as it depends on the weight matrix $w_i$ of layer $i$ can be written as $f_{w_i}(x) = g_i(w_i h_{i-1}(x) + b_i)$. Now consider a small perturbation $x + \delta x$ of the input $x$. There exists a corresponding perturbation $w_i + u(\delta x)$ of the weight matrix in layer $i$ such that

$$f_{w_i}(x + \delta x) = f_{w_i + u(\delta x)}(x). \tag{46}$$

To see this, note that for sufficiently small $\delta x$ we can identify $h_i(x + \delta x)$ with its linear approximation around $x$ so that $h_i(x + \delta x) = h_i(x) + \nabla_x h_i(x)\delta x$, where $\nabla_x h_i : \mathbb{R}^d \to \mathbb{R}^{|h_i|}$ is the total derivative of $h_i : \mathbb{R}^d \to \mathbb{R}^{|h_i|}$. Thus, by representing $f_{w_i}$ as in (42), we have

$$\begin{aligned} f_{w_i}(x + \delta x) &= g_i(w_i h_{i-1}(x + \delta x) + b_i) \\ &= g_i(w_i h_{i-1}(x) + w_i \nabla_x h_{i-1}(x)\delta x + b_i). \end{aligned}$$

Similarly for $f_{w_i + u(\delta x)}(x)$, we have

$$\begin{aligned} f_{w_i + u(\delta x)}(x) &= g_i((w_i + u(\delta x))h_{i-1}(x) + b_i) \\ &= g_i(w_i h_{i-1}(x) + u(\delta x)h_{i-1}(x) + b_i). \end{aligned}$$

Thus Eqn. (46) is satisfied provided $u(\delta x)h_{i-1}(x) = w_i \nabla_x h_{i-1}(\delta x)$. Using the fact that $h_i(x)h_i(x)^T = \|h_i(x)\|_2^2$ and rearranging terms, we get

$$u(\delta x) = \frac{w_i(\nabla_x h_{i-1}(x)\delta x)h_{i-1}(x)^T}{\|h_{i-1}(x)\|_2^2}. \tag{47}$$

Defining $u(\delta x)$ as in Eqn. (47) and taking the derivative of both sides of Eqn. (46) with respect to $\delta x$ at $\delta x = 0$ via the chain rule, we get (since by Eqn. (47) we have $u(\delta x)$ is linear in $\delta x$)

$$\nabla_x f_{w_i}(x) = \nabla_{w_i} f_\theta(x) \frac{(w_i \nabla_x h_{i-1}(x))h_{i-1}^T(x)}{\|h_{i-1}(x)\|_2^2}. \tag{48}$$

Finally, taking the square of the $L^2$ operator norm $\|\cdot\|_2$ on both sides, we have

$$
\begin{aligned}
\|\nabla_x f_{w_i}(x)\|_2^2 &= \left\|\nabla_{w_i} f_\theta(x)\frac{w_i\nabla_x h_{i-1}(x)h_{i-1}^T(x)}{\|h_{i-1}(x)\|_2^2}\right\|_2^2 \\
&\leq \|\nabla_{w_i} f_\theta(x)\|_2^2\frac{\|w_i\|_2^2\|\nabla_x h_{i-1}(x)\|_2^2}{\|h_{i-1}(x)\|_2^2}.
\end{aligned}
$$

Rearranging the terms and since $\nabla_x f_{w_i} = \nabla_x f_\theta$ we get (45) which completes the proof. $\qquad\square$

Following the same argument, we can also prove the corresponding lemma with respect to the derivative of the biases $b_i$ at each layer:

**Lemma A.3.** *Let $f_\theta : \mathbb{R}^d \to \mathbb{R}^k$ represent a deep neural network consisting of $l$ consecutive dense layers. Using the notation above, for $i = 1, 2, \ldots, l$, we have*

$$
\|\nabla_x f_\theta(x)\|_2^2 \left(\frac{1}{\|w_i\|_2\|\nabla_x h_{i-1}(x)\|_2}\right)^2 \leq \|\nabla_{b_i} f_\theta(x)\|_2^2, \tag{49}
$$

*where $\|\cdot\|_2$ denotes the $L^2$ operator norm when applied to matrices and the $L^2$ norm when applied to vectors.*

*Proof.* The proof is similar in spirit to the proof of Lemma A.2. Namely, for small perturbations $x + \delta x$ of the input $x$, we verify that we can find a corresponding small perturbation $b_i + u(\delta x)$ of the bias $b_i$ so that $f_{b_i}(x + \delta x) = f_{b_i + u(\delta x)}(x)$. From Eqn. (42), this time in relation to the bias term, we can write $f_{b_i}(x) = g_i(w_i h_{i-1}(x) + b_i)$. Then, taking a small perturbation $x + \delta x$ of the input $x$ and simplifying as before, we get

$$
u(\delta x) = w_i\nabla_x h_{i-1}(x)\delta x. \tag{50}
$$

As before, taking this as our definition of $u(\delta x)$, and taking the derivative w.r.t. $\delta(x)$ as before via the chain rule, we obtain

$$
\nabla_x f_{b_i}(x) = \nabla_{b_i} f_\theta(x)w_i\nabla_x h_{i-1}(x). \tag{51}
$$

Again, after taking the square of the $L^2$ operator norm on both sides, and moving the terms around we get

$$
\|\nabla_x f_\theta(x)\|_2^2 \left(\frac{1}{\|w_i\|_2\|\nabla_x h_{i-1}(x)\|_2}\right)^2 \leq \|\nabla_{b_i} f_\theta(x)\|_2^2. \tag{52}
$$

$\qquad\square$

It remains to prove Theorem A.1:

*Proof of Theorem A.1.* Recall, that $f_\theta(x)$ represents a deep neural network consisting of $l$ layers and parameterized by $\theta = (w_1, b_1, w_2, b_2, \ldots, w_l, b_l)$. Furthermore, we use the notation $f_{w_i}$ (resp. $f_{b_i}$) to denote $f_\theta$ as dependent only on $w_i$ (resp. $b_i$); i.e., all other parameter values are considered constant. Let $\|\cdot\|_F$ denote the Frobenius norm. For this model structure, by Pythagoras's theorem, it follows that

$$
\|\nabla_\theta f_\theta(x)\|_F^2 = \|\nabla_{w_1} f_\theta(x)\|_F^2 + \|\nabla_{b_1} f_\theta(x)\|_F^2 + \cdots + \|\nabla_{w_l} f_\theta(x)\|_F^2 + \|\nabla_{b_l} f_\theta(x)\|_F^2. \tag{53}
$$

The remainder of the proof relies on the following general property of matrix norms: Given a matrix $A \in \mathbb{R}^{m \times n}$, let $\sigma_{\max}(A)$ represent the largest singular value of $A$. The Frobenius norm $\|\cdot\|_F$ and the $L^2$ operator norm $\|\cdot\|_2$ are related by the following inequalities

$$
\|A\|_2^2 = \sigma_{\max}^2(A) \leq \|A\|_F^2 = \sum_{i=1}^{\min(m,n)} \sigma_k^2 \leq \min(m,n) \cdot \sigma_{\max}^2(A) = \min(m,n) \cdot \|A\|_2^2 \tag{54}
$$

where $\sigma_i(A)$ are the singular values of the matrix $A$. Considering $\nabla_x f_\theta$ as a map from $\mathbb{R}^d$ to $\mathbb{R}^k$, by (54) it follows that

$$
\|\nabla_x f_\theta(x)\|_2^2 \geq \frac{1}{\min(d,k)}\|\nabla_x f_\theta(x)\|_F^2. \tag{55}
$$

By Lemma A.2 and Lemma A.3, we have that for $i = 1, 2, \ldots, l$

$$\|\nabla_{w_i} f_\theta(x)\|_2^2 + \|\nabla_{b_i} f_\theta(x)\|_2^2 \geq \|\nabla_x f_\theta(x)\|_2^2 \left( \frac{1 + \|h_{i-1}(x)\|_2^2}{\|w_i\|_2^2 \|\nabla_x h_{i-1}(x)\|_2^2} \right). \tag{56}$$

Therefore, we can re-write Eqn. (53) using (54), (55) and (56) to get

$$
\begin{aligned}
\|\nabla_\theta f_\theta(x)\|_F^2 &= \sum_{i=1}^{l} \left( \|\nabla_{w_i} f_\theta(x)\|_F^2 + \|\nabla_{b_i} f_\theta(x)\|_F^2 \right) \\
&\geq \sum_{i=1}^{l} \|\nabla_{w_i} f_\theta(x)\|_2^2 + \|\nabla_{b_i} f_\theta(x)\|_2^2 \\
&\geq \sum_{i=1}^{l} \|\nabla_x f_\theta(x)\|_2^2 \left( \frac{1 + \|h_{i-1}(x)\|_2^2}{\|w_i\|_2^2 \|\nabla_x h_{i-1}(x)\|_2^2} \right) \\
&\geq \|\nabla_x f_\theta(x)\|_F^2 \sum_{i=1}^{l} \frac{1}{\min(d,k)} \left( \frac{1 + \|h_{i-1}(x)\|_2^2}{\sigma_{\max}^2(w_i) \sigma_{\max}^2(\nabla_x h_{i-1}(x))} \right).
\end{aligned}
$$

For $i = 1, 2, \ldots, l$, define

$$T_i(x, \theta) := \frac{1}{\sqrt{\min(d,k)}} \frac{\sqrt{1 + \|h_{i-1}(x)\|_2^2}}{\sigma_{\max}(w_i) \sigma_{\max}(\nabla_x h_{i-1}(x))}. \tag{57}$$

We call $T_i(x, \theta)$ the *transfer function for layer $i$* in the network.

Then, rearranging terms in the final inequality above, we get

$$\|\nabla_x f_\theta(x)\|_F^2 \leq \frac{\|\nabla_\theta f_\theta(x)\|_F^2}{T_1^2(x, \theta) + \cdots + T_l^2(x, \theta)}. \tag{58}$$

This completes the proof. $\qquad\square$

# B  Experiment details

## B.1  Figure 1: Motivating 1D example

We trained a ReLU MLP with 3 layers of 300 neurons each to regress 10 data points on an exact parabola $(x, x^2) \in \mathbb{R}^2$ where $x$ ranges over 10 equidistant points in the interval $[-1, 1]$. We use full batch gradient descent with learning rate 0.02 for 100000 steps/epochs. We plotted the model function across the $[-1, 1]$ range and computed its geometric complexity over the dataset at step 0 (initialization), step 10, step 1000, and step 10000 (close to interpolation). The network was randomly initialized using the standard initialization (i.e. the weights were sampled from a truncated normal distribution with variance inversely proportional to the number of input units and the bias terms were set to zero). This model was trained five separate times using five different random seeds. Each line marks the mean of the five runs and the shaded region is 95% confidence interval over these five seeds.

## B.2  Figure 2: Geometric complexity and initialization

**Left and Middle:**  We initialized several ReLU MLP's $f_{\theta_0} : \mathbb{R}^d \to \mathbb{R}^k$ with large input and output: $d = 224 \times 224 \times 3$ and $k = 1000$, with 500 neurons per layer, and with a varying number of layers $l \in [1, 2, 4, 8, 16, 32, 64]$. We used the standard initialization scheme: we sample them from a truncated normal distribution with variance inversely proportional to the number of input units and the bias terms were set to zero. We measured and plotted the following quantities

$$
\begin{aligned}
f_{\text{mean}}(x) &= \text{mean} \left\{ f_{\theta_0}^i(P_1 + (P_2 - P_1)x), \quad i = 1, \ldots, 1000 \right\}, \tag{59} \\
f_{\text{max}}(x) &= \text{max} \left\{ (|f_{\theta_0}^i(P_1 + (P_2 - P_1)x)|, \quad i = 1, \ldots, 1000 \right\}, \tag{60}
\end{aligned}
$$

with $x$ ranging over 50 equidistant points in the interval $[0, 1]$. The points $P_1$ and $P_2$ where chosen to be the two diagonal points $(-1, \ldots, -1)$ and $(1, \ldots, 1)$, respectively, of the normalized data

hyper-cube $[-1, 1]^d$. Each line marks the mean of the 5 runs and the shaded region is 95% confidence interval over these five seeds.

What we observe with the standard initialization scheme also persists with the Glorot initialization scheme, where the biases are set to zero, and the weight matrices parameters are sampled from the uniform distribution on $[-1, 1]$ and scaled at each layer by $\sqrt{6/d_l + d_{l-1}}$, where $d_l$ is the number of units in layer $l$. We report this in Fig. 6 below.

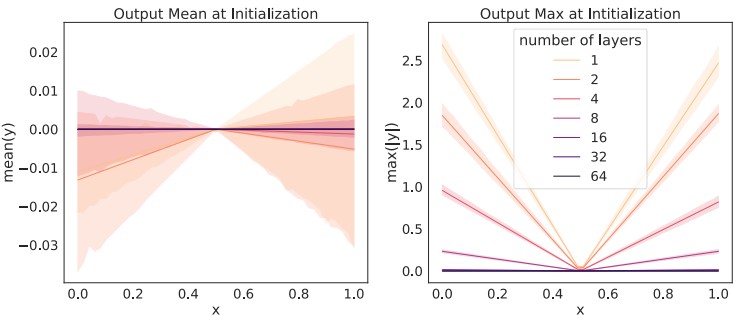

Figure 6: Additional plots to complement Fig. 2: Deeper neural Relu networks initialize closer to the zero function with the Glorot scheme on normalized data. We repeat the setup of Fig. 2 Left and Middle described in Section B.2 but with the Glorot scheme instead of the standard scheme.

**Right:** We measured the geometric complexity $\langle f_{\theta_0}, D \rangle_G$ at initialization for ReLU MLP's $f_{\theta_0} : \mathbb{R}^d \to \mathbb{R}^k$ with large input and output: $d = 224 \times 224 \times 3$ and $k = 1000$, with 500 neurons per layer, and with a varying number of layers $l \in [1, 2, 4, 8, 16, 32, 64]$. The geometric complexity was computed over a dataset $D$ of 100 points sampled uniformly from the normalized data hyper-cube $[-1, 1]^d$. For each combination of activation and initialization in $\{\text{ReLU}, \text{sigmoid}\} \times \{\text{standard}, \text{Glorot}\}$ we repeated the experiment 5 times with different random seeds. The Glorot initialization scheme is the one described in the paragraph above. For the standard initialization we set the biases to zero, and initialized the weight matrices parameters from a normal distribution truncated to the range $[-2, 2]$ and rescaled using $1/\sqrt{d_{l-1}}$ at each layer. We plotted the mean geometric complexity with error bars representing the 95% confidence interval over the 5 random seeds. The error bars are tiny in comparison of the plotted quantities and are therefore not visible on the plot.

### B.3 Figure 3: Geometric complexity and explicit regularization

**Left:** We trained a ResNet18 on CIFAR10 three times with different random seeds with a learning rate of 0.02, batch size of 512, for 10000 steps for each combination of regularization rate and regularization type in $\{\text{L2}, \text{Spectral}, \text{Flatness}\} \times [0, 0.01, 0.025, 0.05, 0.075, 0.1]$. We measured the geometric complexity at time of maximum test accuracy for each of these runs and plotted the mean with 95% confidence interval over the 3 random seeds. For the L2 regularization we added the sum of the parameter squares to the loss multiplied with the regularization rate. For the spectral regularization, we followed the procedure described in [96] by adding to the loss the penalty $(\alpha/2) \sum_i \sigma_{\max}(W_i)^2$ where $W_i$ is either the layer weight matrix for a dense layer or, for a convolution layer, the matrix of shape $b \times a k_w k_h$ obtained by reshaping the convolution layer with $a$ input channels, $b$ output channels, and a kernel of size $k_w \times k_h$. For the flatness regularization, we added to the batch loss $L_B$ at each step the norm square of the batch loss gradient $\|\nabla_\theta L_B(\theta)\|^2$ multiplied by the regularization rate.

**Middle:** We trained a ResNet18 on CIFAR10 with learning rate 0.005 and batch size 16 for 100000 steps with varying proportion of label noise. For each label noise proportion $\alpha \in [0, 0.05, 0.1, 0.15, 0.25]$, we trained three times with different random seeds. We plotted the mean geometric complexity at time of maximum test accuracy as well as the 95% confidence interval over the three runs. The label noise was created by mislabelling $\alpha$ % of the true labels before training.

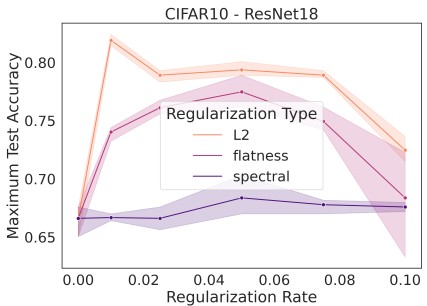

Figure 7: Additional plot to complement Fig. 3: Maximum test accuracy recorded for different levels of explicit regularization.

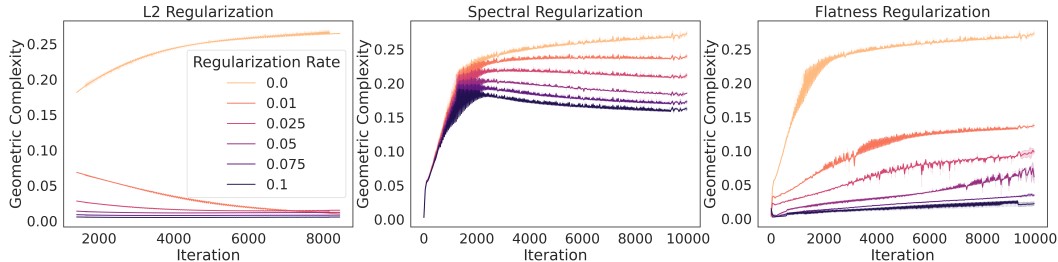

Figure 8: Additional plots to complement Fig. 3: GC plotted against training iterations for different explicit regularization types and rates for ResNet18 trained on CIFAR10.

**Right:** We trained a MLP with six layers of 214 neurons on MNIST with learning rate 0.005 and batch size 32 for 128000 steps with varying proportion of label noise. For each label noise proportion $\alpha \in [0, 0.05, 0.1, 0.15, 0.25]$, we trained the neural network 5 times with a different random seed. We plotted the mean geometric complexity at time of maximum accuracy as well as the 95% confidence interval over the 5 runs. The label noise was created by mislabelling $\alpha$ % of the true labels before training.

### B.4 Figure 4: Geometric complexity and implicit regularization

**Top row (varying learning rates):** We trained a ResNet18 on CIFAR10 with batch size 512, for 30000 steps with varying learning rates. For each learning rate in $[0.005, 0.01, 0.05, 0.1, 0.2]$, we trained the neural network three times with a different random seed. We plotted the learning curves for the test accuracy, the geometric complexity, and the training loss, where the solid lines represent the mean of these quantities over the three runs and the shaded area represents the 95% confidence interval over the three runs. We applied a smoothing over 50 steps for each run, before computing the mean and the confidence interval of the three runs.

**Bottom row (varying batch sizes):** We trained a ResNet18 on CIFAR10 with learning rate 0.2, for 100000 steps with varying batch sizes. For each batch size in $[8, 16, 32, 64, 128, 256, 512, 1024]$, we trained the neural network three times with a different random seed. We plotted the learning curves for the test accuracy, the geometric complexity, and the training loss, where the solid lines represent the mean of these quantities over the three runs and the shaded area represents the 95% confidence interval over the three runs. We applied a smoothing over 50 steps for each run, before computing the mean and the confidence interval of the three run.

### B.5 Figure 5: Geometric complexity and double-descent

We train a ResNet18 on CIFAR10 with learning rate 0.8, batch size 124, for 100000 steps, with 18 different network widths. For each network width in $[1, 2, 3, 4, 5, 6, 7, 8, 9, 10, 11, 12, 13, 14, 20, 30, 64, 100, 128]$ we train three times with a different random seed.

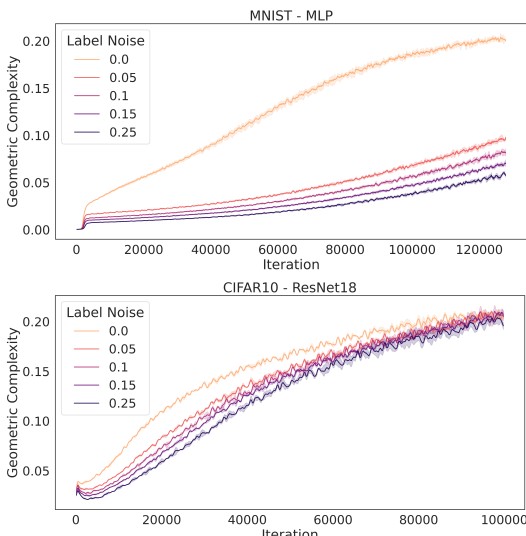

Figure 9: Additional plots to Fig. 3 experiments: GC plotted against training iterations for different label noise proportions.

**Left:**  We measure the geometric complexity as well as the test loss at the end of training. We plot the mean and the 95% confidence interval of both quantities over the three runs against the network width. The critical region in yellow identified experimentally in [75] indicates the transition between the under-parameterized regime and the over-parameterized regime, where the second descent starts.

**Right:**  We plot the test loss measured at the end of training against the geometric complexity at the end of training. The top plot shows every model width for every seed, while the bottom plot shows averages of these quantities over the three seeds. We then fit the data with a polynomial of degree six to sufficiently capture any high order relationship between the test loss and geometric complexity.

**Definition of width:**  We follow the description of ResNet width discussed in [75]. The ResNet18 architecture we follow is that of [42]. The original ResNet18 architecture has four successive ResNet blocks each formed by two identical stacked sequences of a BatchNorm and a convolution layer. The number of filters for the convolution layers in each of the successive ResNet block is $(k, 2k, 4k, 8k)$ with $k = 64$ for the original RestNet18. Following [75], we take $k$ to be the width of the network, which we vary during our experiments.

# C Additional Experiments

## C.1 Geometric complexity at initialization decreases with added layers on large domains

We reproduce the experiments from Fig. 2 on a larger domain with the same conclusion: For both the standard and the Glorot initialization schemes ReLU networks initialize closer to the zero function and with lower geometric complexity as the number of layer increases.

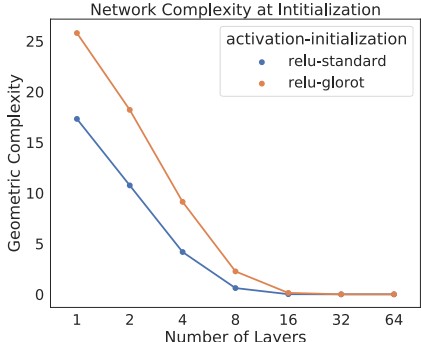

Figure 10: **Geometric complexity at initialization decreases with the number of layers up to zero even on large domains:** We repeat the setup of Fig. 2 Right described in Section B.2 but we sample the dataset from points in a large domain $[-1000, 1000]^d$ instead of the normalized hyper-cube $[-1, 1]^d$. We measure the geometric complexity for both the Glorot and the standard initialization schemes.

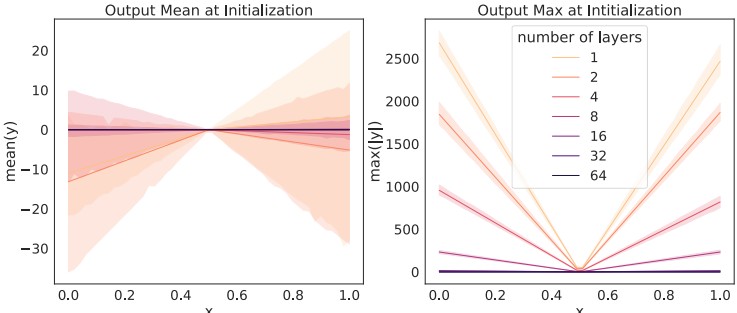

Figure 11: **Deeper neural Relu networks initialize closer to the zero function with the Glorot scheme even on large domains:** We repeat the setup of Fig. 2 Left and Middle described in Section B.2 but we evaluate the networks on a diagonal of the larger hyper-cube $[-1000, 1000]^d$ instead of the normalized hyper-cube $[-1, 1]^d$.

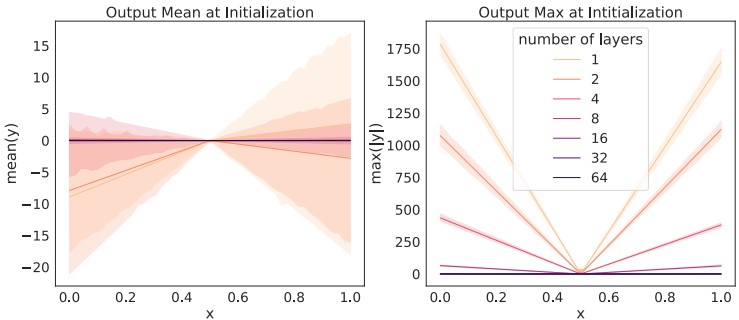

Figure 12: **Deeper neural Relu networks initialize closer to the zero function with the standard scheme even on large domains:** We repeat the setup of Fig. 2 Left and Middle described in Section B.2 but we evaluate the networks on a diagonal of the larger hyper-cube $[-1000, 1000]^d$ instead of the normalized hyper-cube $[-1, 1]^d$.

## C.2 Geometric complexity decreases with implicit regularization for MNIST

We replicate the implicit regularization experiments we conducted for CIFAR10 with ResNet18 in Fig. 4 for MLP's trained on MNIST. The conclusion is the same: higher learning rates and smaller batch sizes decrease geometric complexity through implicit gradient regularization, and are correlated with higher test accuracy.

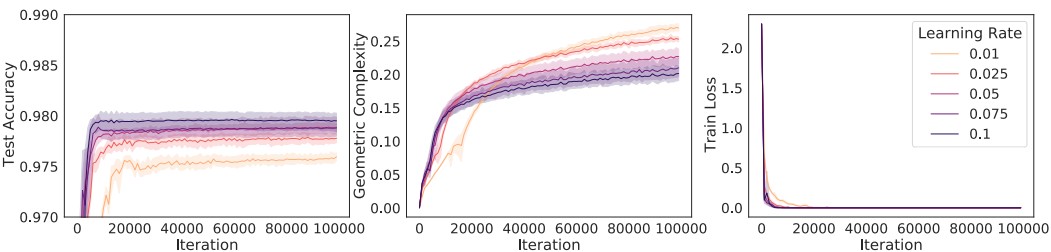

Figure 13: **Geometric complexity decreases with higher learning rates on MNIST**: We trained a selection of MLP's with 6 hidden layers with 500 neurons per layer on MNIST with batch size of 512, for 100000 steps and with varying batch sizes. For each learning rate in $[0.01, 0.025, 0.05, 0.075, 0.1]$, we trained 5 different times with a different random seed. The MLP were initialized using the standard initialization scheme.

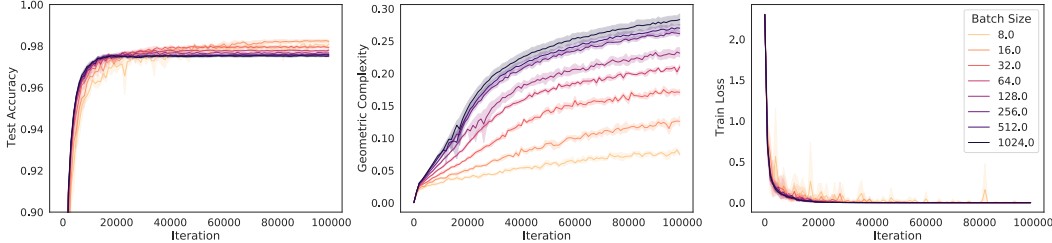

Figure 14: **Geometric complexity decreases with smaller batch sizes on MNIST**: We trained a selection of MLP's with 6 hidden layers with 500 neurons per layer on MNIST with learning rate of 0.02, for 100000 steps and with varying batch sizes. For each batch size in $[8, 16, 32, 64, 128, 256, 512, 1024]$, we trained 5 different times with a different random seed. The MLP were initialized using the standard initialization scheme.

## C.3 Geometric complexity decreases with explicit regularization for MNIST

We replicate the explicit regularization experiments we conducted for CIFAR10 with ResNet18 in Fig. 3 for MLP's trained on MNIST with similar conclusions: higher regularization rates for L2, spectral, and flatness regularization decrease geometric complexity and are correlated with higher test accuracy.

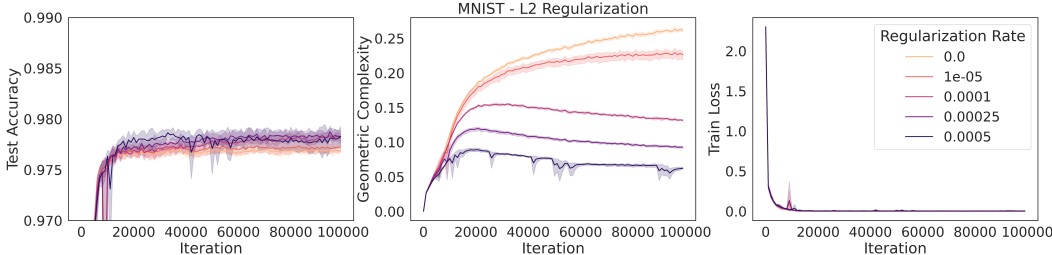

Figure 15: **Geometric complexity decreases with L2 regularization on MNIST**: We trained a selection of MLP's with 6 hidden layers with 500 neurons per layer on MNIST with learning rate of 0.02, batch size of 512, for 100000 steps. We regularized the loss by adding to it the L2 norm penalty $\alpha \sum_i \|W_i\|_F^2$ where $W_i$ are the layer weight matrices. For each regularization rate $\alpha \in [0, 0.00001, 0.0001, 0.00025, 0.0005]$, we trained 5 different times with a different random seed. The MLP were initialized using the standard initialization scheme.

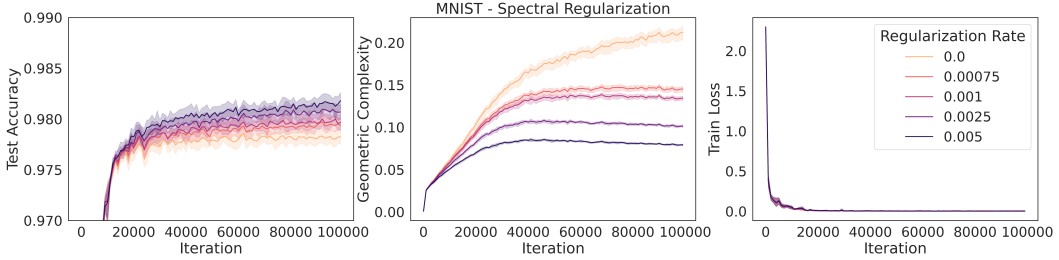

Figure 16: **Geometric complexity decreases with spectral regularization on MNIST**: We trained a selection of MLP's with 4 hidden layers with 200 neurons per layer on MNIST with learning rate of 0.02, batch size of 128, for 100000 steps. We regularized the loss by adding to it the spectral norm penalty $\alpha/2 \sum_i \sigma_{\max}(W_i)^2$ where $W_i$ are the layer weight matrices as described in [71]. For each regularization rate $\alpha \in [0, 0.00075, 0.001, 0.0025, 0.005]$, we trained 5 different times with a different random seed. The MLP were initialized using the standard initialization scheme.

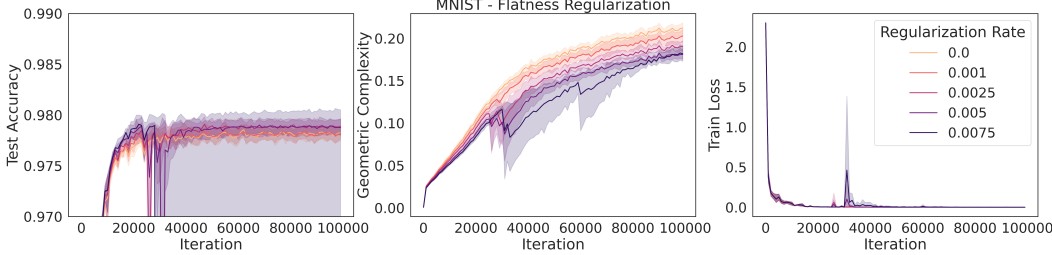

Figure 17: **Geometric complexity decreases with flatness regularization on MNIST**: We trained a selection of MLP's with 4 hidden layers with 200 neurons per layer on MNIST with learning rate of 0.02, batch size of 128, for 100000 steps. We regularized the loss by adding to it the gradient penalty $\alpha\|\nabla_\theta L_B(\theta)\|^2$ where $L_B$ is the batch loss. For each regularization rate $\alpha \in [0, 0.001, 0.0025, 0.005, 0.0075]$, we trained 5 different times with a different random seed. The MLP were initialized using the standard initialization scheme.

## C.4 Explicit geometric complexity regularization for MNIST and CIFAR10

In this section, we explicitly regularize for the geometric complexity. This is a known form of regularization also called Jacobian regularization [46, 90, 95, 96]. We first perform this regularization for a MLP trained on MNIST (Fig. 18) and then for a ResNet18 trained on CIFAR10 (Fig. 19) with the following conclusion: test accuracy increases with higher regularization strength while the geometric complexity decreases.

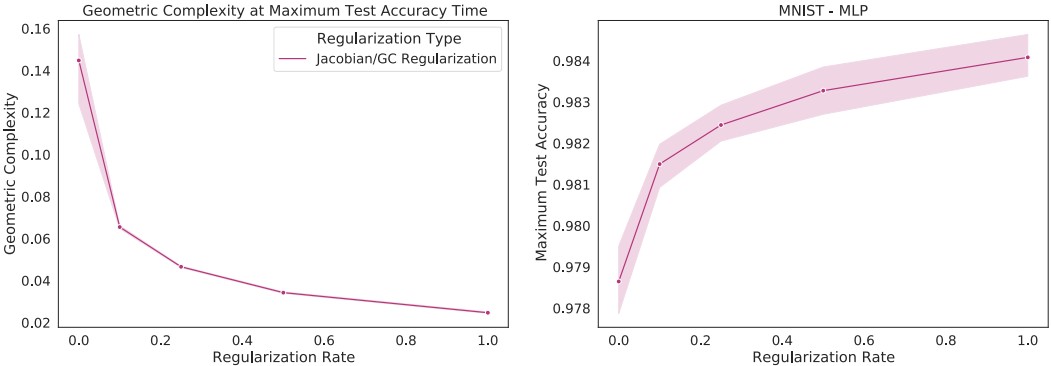

Figure 18: **Test accuracy increases with explicit GC regularization on MNIST**: We trained a selection of MLP's with 4 hidden layers with 200 neurons per layer on MNIST with learning rate of 0.02, batch size of 128, for 100000 steps. We regularized the loss by adding to it the gradient penalty $\alpha/B \sum_{x \in B} \|\nabla_x f_\theta(x)\|_F^2$ where $f_\theta(x)$ is the logit network. For each regularization rate $\alpha \in [0, 0.1, 0.25, 0.5, 1]$, we trained 5 different times with a different random seed. The MLP were initialized using the standard initialization scheme.

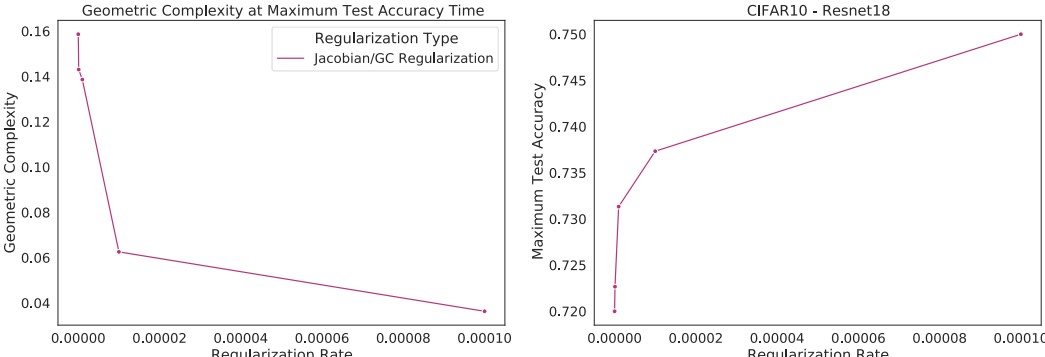

Figure 19: **Test accuracy increases with explicit GC regularization on CIFAR10**: We trained a selection of ResNet18 with learning rate of 0.02, batch size of 128, for 10000 steps. We regularized the loss by adding to it the gradient penalty $\alpha/B \sum_{x \in B} \|\nabla_x f_\theta(x)\|_F^2$ where $f_\theta(x)$ is the logit network. For each regularization rate $\alpha \in [0, 0.0000001, 0.000001, 0.00001, 0.0001]$, we trained only one time with a single random seed, and the training had to be stopped before reaching peak test accuracy because of the heavy computational time due to this regularization.

## C.5 Separate L2, flatness, and spectral regularization experiments for CIFAR10

For the sake of space in Fig. 3 (right) in the main paper, we used the same regularization rate range for all types of explicit regularization we tried. In this section, we perform the experiments on a targeted range for each regularization type, leading to clearer plots (Fig. 20, Fig. 21, and Fig. 22).

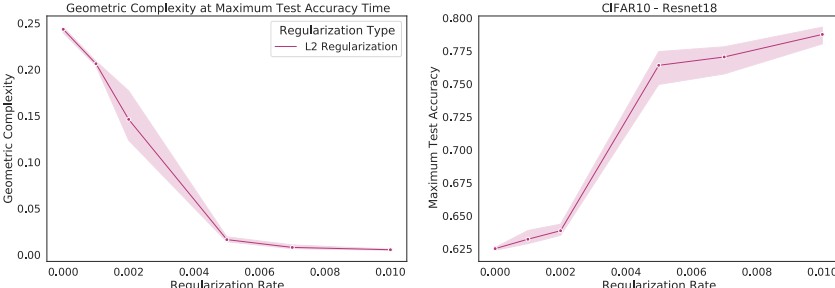

Figure 20: **GC decreases with explicit L2 regularization on CIFAR10**: We trained a selection of ResNet18 with learning rate of 0.02, batch size of 512, for 10000 steps. We regularized the loss by adding to it the standard L2 loss penalty. For each regularization rate $\alpha \in [0, 0.001, 0.002, 0.005, 0.007, 0.01]$, we trained 3 different times with a different random seed.

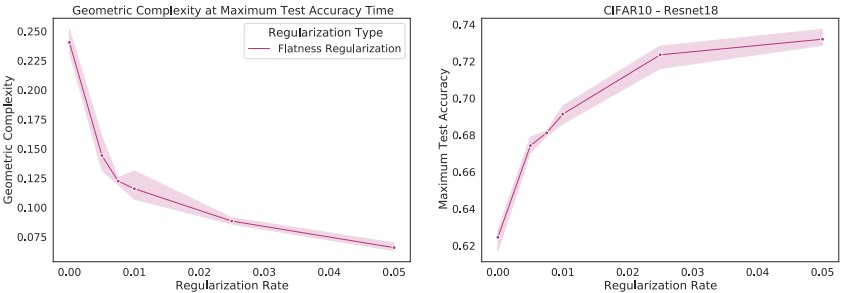

Figure 21: **GC decreases with explicit flatness regularization on CIFAR10**: We trained a selection of ResNet18 with learning rate of 0.02, batch size of 512, for 10000 steps. We regularized the loss by adding to it the gradient penalty $\alpha \|\nabla_\theta L_B(\theta)\|^2$ where $L_B$ is the batch loss. For each regularization rate $\alpha \in [0, 0.005, 0.0075, 0.01, 0.025, 0.05]$, we trained 3 different times with a different random seed.

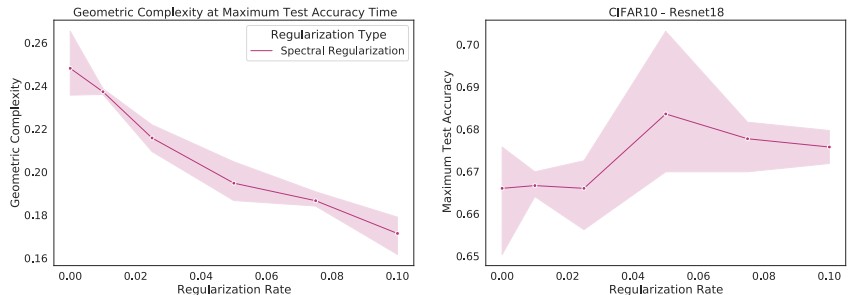

Figure 22: **Geometric complexity decreases with spectral regularization on CIFAR10:** We trained a selection of ResNet18 on CIFAR10 with learning rate of 0.02, batch size of 512, for 100000 steps. For each regularization rate $\alpha \in [0, 0.01, 0.025, 0.05, 0.075, 0.1]$, we trained 3 different times with a different random seed.

## C.6 Geometric complexity in the presence of multiple tuning mechanisms

For most of this paper, we studied the impact of tuning strategies, like the choice of initialization, hyper-parameters, or explicit regularization in isolation from other very common heuristics like learning rate schedules and data-augmentation. In this section, we reproduce the implicit regularization effect of the batch size and the learning rate on GC (c.f. Fig. 4 in the main paper) while using these standard tricks to achieve better performance. The resulting models achieve performance closer to SOTA for the ResNet18 architecture.

Although the learning curves are messier and harder to interpret (Fig. 23) because of the multiple mechanisms interacting in complex ways, we still observe that the general effect of the learning rate (Fig. 24) and the batch size (Fig. 25) on geometric complexity is preserved in this context. More importantly, we also note that the sweeps with higher test accuracy solutions tend also to come with a lower geometric complexity, even in this more complex setting (Fig. 24 right and Fig. 25 right). Namely, models with higher test accuracy have correspondingly lower GC. Specifically in terms of implicit regularization, as the learning rate increases, the geometric complexity decreases and the maximum test accuracy increases. Similarly, smaller batch size leads to lower geometric complexity as well as higher test accuracy.

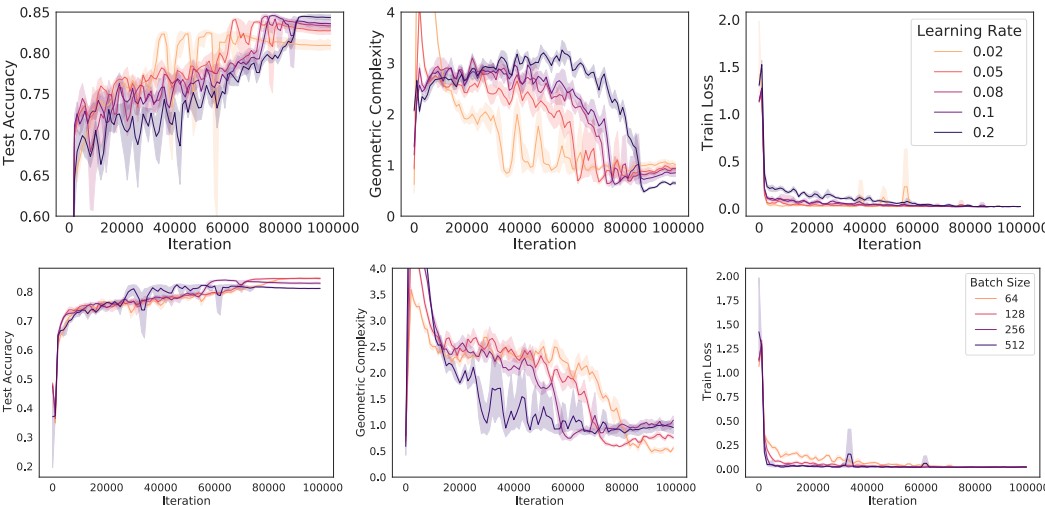

Figure 23: Impact of IGR when training ResNet18 on CIFAR10 with multiple tuning mechanisms including cosine learning rate scheduler, data augmentation and L2 regularization. Note that the GC is computed on batches of size 128 which leads to a lot of variance in the estimate. **Top row:** As IGR increases through higher learning rates, GC decreases. **Bottom row:** Similarly, lower batch size leads to decreased GC.

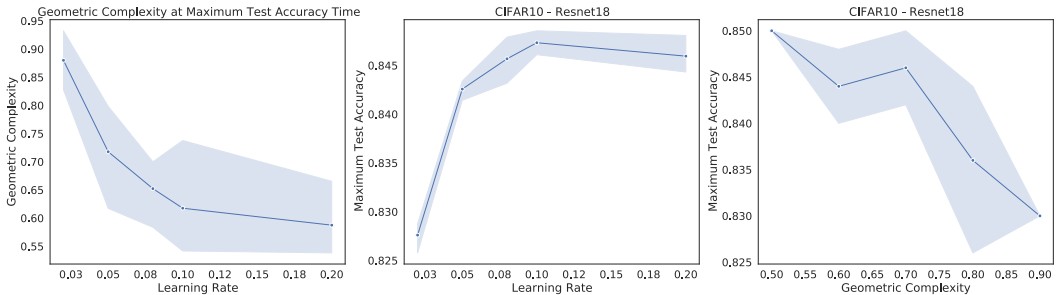

Figure 24: **GC decreases as learning rate and model test accuracy increases on CIFAR10**: We trained a collection of ResNet18 models on CIFAR10 with varying initial learning rates $h \in [0.02, 0.05, 0.08, 0.1, 0.2]$ and cosine learning rate schedule. Each job was trained with SGD without momentum for 100000 steps, with batch size 128 and L2 regularized loss with regularization rate 0.005. We also included data augmentation in the form of random flip. Test accuracy is reported as best test accuracy during training. GC is computed during training on the training batches, which produces a large variance in the estimate when the batch size is small.

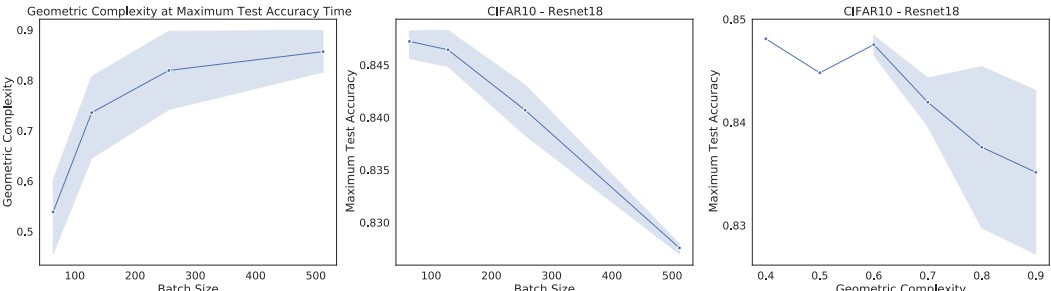

Figure 25: **GC increases as batch size increases on CIFAR10**: We trained a collection of ResNet18 models on CIFAR10 with varying batch sizes of 64, 128, 256, and 512. Each job was trained with SGD without momentum for 100000 steps, with cosine learning rate scheduler initialized at 0.02 and L2 regularized loss with regularization rate 0.005. We also included data augmentation in the form of random flip. Test accuracy is reported as best test accuracy during training. GC is computed during training on the training batches, which produces a large variance in the estimate when the batch size is small.

## C.7 Geometric complexity in the presence of momentum

We replicate the implicit and explicit regularization experiments using SGD *with momentum*, which is widely used in practise. The conclusion remains the same as for vanilla SGD: More regularization (implicit through batch size or learning rate or explicit through flatness, spectral, and L2 penalties) produces solutions with higher test accuracy and lower geometric complexity.

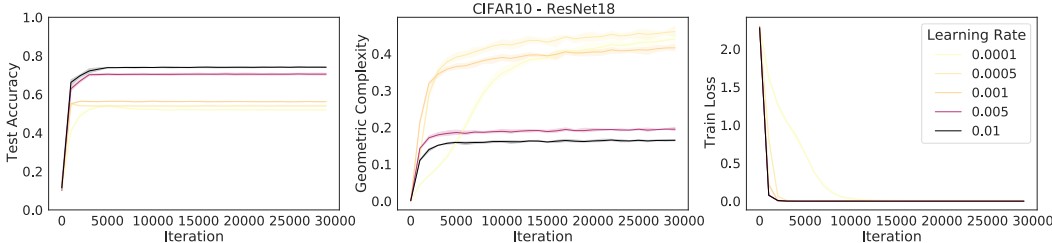

Figure 26: **Geometric complexity decreases with higher learning rates on CIFAR10 trained with momentum**: We trained a selection of ResNet18 with batch size of 512 for 30000 steps using SGD with a momentum of 0.9. For each learning rate $\alpha \in [0.0001, 0.0005, 0.001, 0.005, 0.01]$, we trained 3 different times with a different random seed.

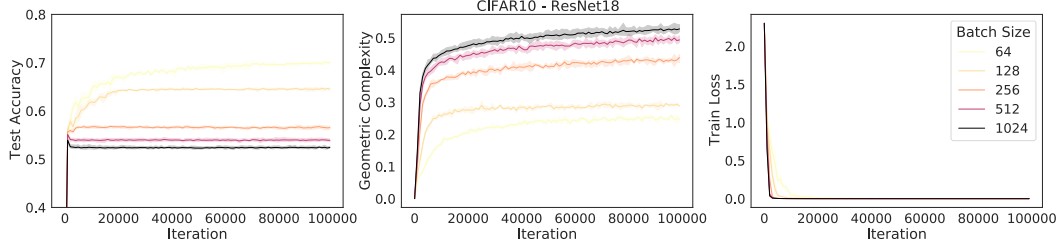

Figure 27: **Geometric complexity decreases with lower batch sizes on on CIFAR10 trained with momentum**: We trained a selection of ResNet18 with learning rate of 0.0005 for 100000 steps using SGD with a momentum of 0.9. For each batch size in $\alpha \in [64, 128, 256, 512, 1024]$, we trained 3 different times with a different random seed.

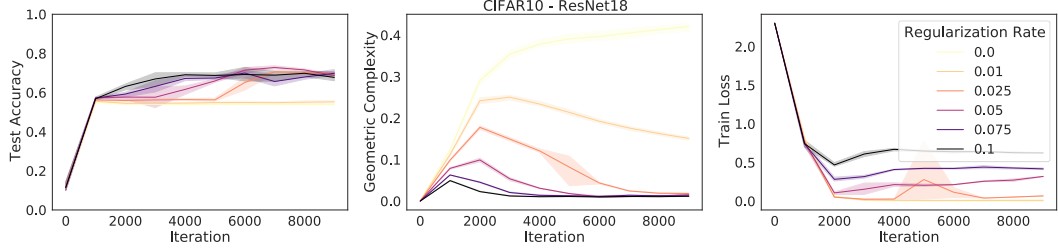

Figure 28: **Geometric complexity decreases with increased L2 regularization on CIFAR10 trained with momentum**: We trained a selection of ResNet18 with learning rate of 0.0005 with batch size of 512 for 10000 steps using SGD with a momentum of 0.9. For each regularization rate in $\alpha \in [0, 0.01, 0.025, 0.05, 0.075, 0.1]$, we trained 3 different times with a different random seed.

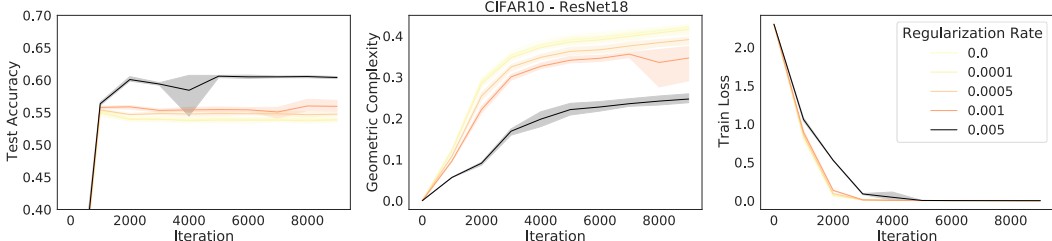

Figure 29: **Geometric complexity decreases with increased flatness regularization on CIFAR10 trained with momentum**: We trained a selection of ResNet18 with learning rate of 0.0005 with batch size of 512 for 10000 steps using SGD with a momentum of 0.9. We regularized the loss by adding to it the gradient penalty $\alpha\|\nabla_\theta L_B(\theta)\|^2$ where $L_B$ is the batch loss. For each regularization rate in $\alpha \in [0, 0.0001, 0.0005, 0.001, 0.005]$, we trained 3 different times with a different random seed.

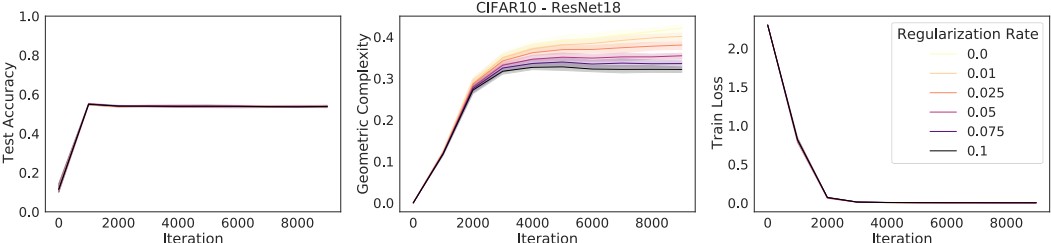

Figure 30: **Geometric complexity decreases with increased spectral regularization on CIFAR10 trained with momentum**: We trained a selection of ResNet18 with learning rate of 0.0005 with batch size of 512 for 10000 steps using SGD with a momentum of 0.9. We regularized the loss by adding to it the spectral norm penalty $\alpha/2 \sum_i \sigma_{\max}(W_i)^2$ where $W_i$ are the layer weight matrices as described in [71]. For each regularization rate in $\alpha \in [0, 0.01, 0.025, 0.05, 0.075, 0.1]$, we trained 3 different times with a different random seed.

## C.8 Geometric complexity in the presence of Adam

We replicate the implicit and explicit regularization experiment using Adam, which is widely used in practice. In this case the conclusions are less clear than with vanilla SGD or SGD with momentum. While higher learning rates, and higher explicit flatness, L2, and spectral regularization still put a regularizing pressure on the geometric complexity for most of the training, the effect of batch size on geometric complexity is not clear. This may be that the local built-in re-scaling of the gradient sizes in Adam affects the pressure on the geometric complexity in complex ways when the batch size changes.

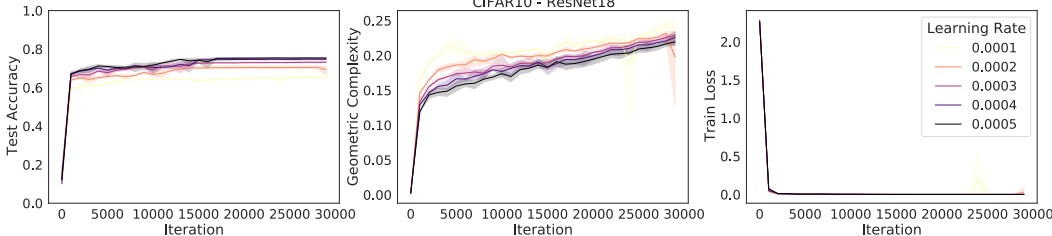

Figure 31: **Geometric complexity decreases with higher learning rates on CIFAR10 trained using Adam with** $b1 = 0.9$**,** $b2 = 0.999$: We trained a selection of ResNet18 with batch size of 512 for 30000 steps using Adam with a momentum of 0.9. For each learning rate $\alpha \in [0.0001, 0.0002, 0.0003, 0.0004, 0.0005]$, we trained 3 different times with a different random seed.

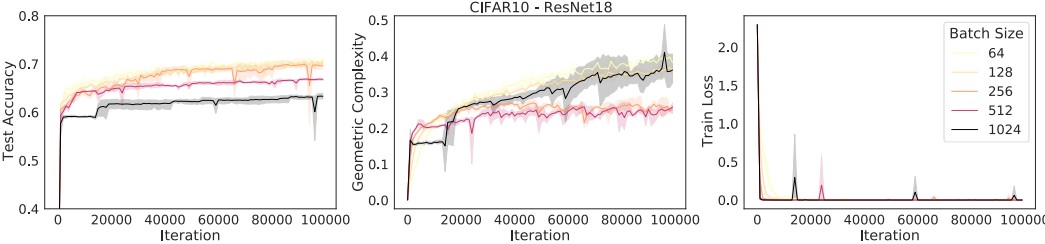

Figure 32: **The relation between geometric complexity and batch size is ambiguous on CIFAR10 trained with Adam**: We trained a selection of ResNet18 with learning rate of 0.0001 for 100000 steps using using Adam with $b1 = 0.9$, $b2 = 0.999$. For each batch size in $\alpha \in [64, 128, 256, 512, 1024]$, we trained 3 different times with a different random seed.

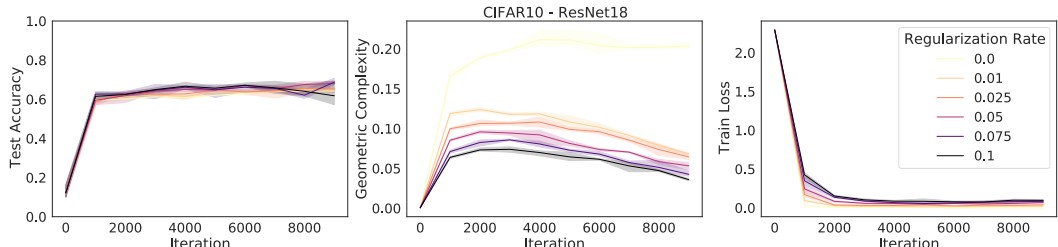

Figure 33: **Geometric complexity decreases with increased L2 regularization on CIFAR10 trained with Adam**: We trained a selection of ResNet18 with learning rate of 0.0001 with batch size of 512 for 10000 steps using using Adam with $b1 = 0.9$, $b2 = 0.999$. For each regularization rate in $\alpha \in [0, 0.01, 0.025, 0.05, 0.075, 0.1]$, we trained 3 different times with a different random seed.

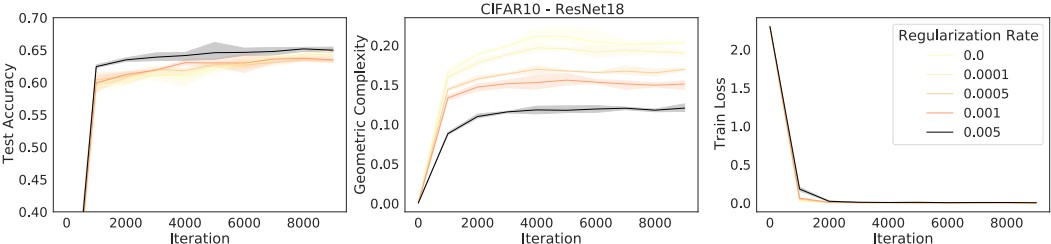

Figure 34: **Geometric complexity decreases with increased flatness regularization on CIFAR10 trained with Adam**: We trained a selection of ResNet18 with learning rate of 0.0001 with batch size of 512 for 10000 steps using using Adam with $b1 = 0.9$, $b2 = 0.999$. We regularized the loss by adding to it the gradient penalty $\alpha \|\nabla_\theta L_B(\theta)\|^2$ where $L_B$ is the batch loss. For each regularization rate in $\alpha \in [0, 0.0001, 0.0005, 0.001, 0.005]$, we trained 3 different times with a different random seed.

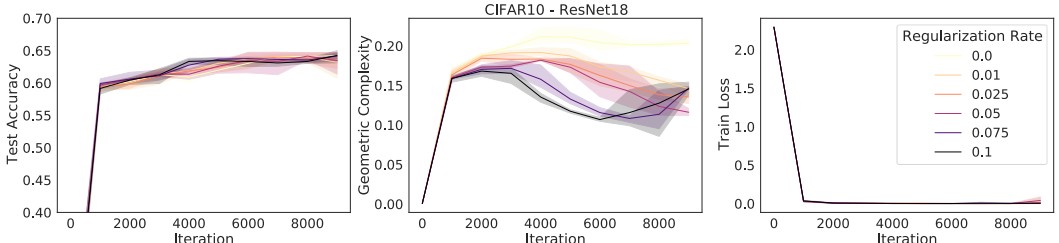

Figure 35: **Geometric complexity decreases with increased spectral regularization on CIFAR10 trained with Adam**: We trained a selection of ResNet18 with learning rate of 0.0001 with batch size of 512 for 10000 steps using Adam with $b1 = 0.9$, $b2 = 0.999$. We regularized the loss by adding to it the spectral norm penalty $\alpha/2 \sum_i \sigma_{\max}(W_i)^2$ where $W_i$ are the layer weight matrices as described in [71]. For each regularization rate in $\alpha \in [0, 0.01, 0.025, 0.05, 0.075, 0.1]$, we trained 3 different times with a different random seed.

## D   Comparison of the Geometric Complexity to other complexity measures

One of the primary challenges in deep learning is to better understand mechanisms or techniques that correlate well with (or can imply a bound on) the generalization error for large classes of models. The standard approach of splitting the data into a train, validation and test set has become the de facto way to achieve such a bound. With this goal in mind, a number of complexity measures have been proposed in the literature with varying degrees of theoretical justification and/or empirical success. In this section we compare our geometric complexity measure with other, more familiar complexity measures such as the Rademacher complexity, VC dimension and sharpness-based measures.

### D.1   Rademacher Complexity

Perhaps the most historically popular and widely known complexity measure is the Rademacher complexity [8, 11, 55, 56]. Loosely speaking the Rademacher complexity measures the degree to which a class of functions $\mathcal{H}$ can fit random noise. The idea behind this complexity measure is that a more complex function space is able to generate more complex representation vectors and thus, on average, produce learned functions that are better able to correlate with random noise than a less complex function space.

To make this definition more precise and frame it in the context of machine learning (see also [72]), given an input feature space $X$ and a target space $Y$, let $\mathcal{G}$ denote a family of loss functions $L : \mathcal{Z} = X \times Y \to \mathbb{R}$ associated with a function class $\mathcal{H}$. Notationally,

$$\mathcal{G} = \{g : (x, y) \mapsto L(h(x), y) : h \in \mathcal{H}\}.$$

We define the *empirical Rademacher complexity* as follows:

**Definition D.1** (Empirical Rademacher Complexity). *With $\mathcal{G}$ as above, let $S = \{z_1, z_2, \ldots, z_m\}$ be a fixed sample of size $m$ of elements of $\mathcal{Z} = X \times Y$. The empirical Rademacher complexity of $\mathcal{G}$ with respect to the sample $S$ is defined as:*

$$\widehat{\mathfrak{R}}_S(\mathcal{G}) = \mathbb{E}_{\boldsymbol{\sigma}} \left[ \sup_{g \in \mathcal{G}} \frac{1}{m} \sum_{i=1}^{m} \sigma_i g(z_i) \right],$$

*where $\boldsymbol{\sigma} = (\sigma_1, \sigma_2, \ldots, \sigma_m)^\intercal$, with the $\sigma_i$'s being independent uniform random variables which take values in $\{-1, +1\}$. These random variables $\sigma_i$ are called* Rademacher variables.

If we let $g_S$ denote the vector of values taken by function $g$ over the sample $S$, then the Rademacher complexity, in essence, measures the expected value of the supremum of the correlation of $g_S$ with a vector of random noise $\boldsymbol{\sigma}$; i.e., the empirical Rademacher complexity measures on average how well the function class $\mathcal{G}$ correlates with random noise on the set $S$. More complex families $\mathcal{G}$ can generate more vectors $g_S$ and thus better correlate with random noise on average, see [72] for more details.

Note that the empirical Rademacher complexity depends on the sample $S$. The Rademacher complexity is then an average of this empirical measure over the distribution from which all samples are drawn:

**Definition D.2** (Rademacher Complexity). *Let $\mathcal{D}$ denote the distribution from which all samples $S$ are drawn. For any integer $m \geq 1$, the* Rademacher complexity *of $\mathcal{G}$ is the expectation of the empirical Rademacher complexity over all samples of size $m$ drawn according to $\mathcal{D}$:*

$$\mathfrak{R}_m(\mathcal{G}) = \mathbb{E}_{S \sim \mathcal{D}^m}[\widehat{\mathfrak{R}}_S(\mathcal{G})].$$

The Rademacher complexity is distribution dependent and defined for any class of real-valued functions. However, computing it can be intractable for modern day machine learning models. Similar to the empirical Rademacher complexity, the geometric complexity is also computed over a sample of points, in this case the training dataset, and is well-defined for any class of differentiable functions. In contrast, the Rademacher complexity (and the VC dimension which we discuss below) measures the complexity for an entire hypothesis space, while the geometric complexity focuses only on single functions. Furthermore, since the Geometric Complexity relies only on first derivatives of the learned model function making it much easier to compute.

## D.2   VC dimension

The Vapnik–Chervonenkis (VC) dimension [16, 21, 94] is another common approach to measuring the complexity of a class of functions $\mathcal{H}$ and is often easier to compute than the Rademacher Complexity, see [72] for further discussion on explicit bounds which compare the Rademacher complexity with the VC dimension.

The VC dimension is a purely combinatorial notion and defined using the concept of a *shattering* of a set of points. A set of points is said to be shattered by $\mathcal{H}$ if, no matter how we assign a binary label to each point, there exists a member of $\mathcal{H}$ that can perfectly separate the points; i.e., the growth function for $\mathcal{H}$ is $2^m$. The VC dimension of a class $\mathcal{H}$ is the size of the largest set that can be shattered by $\mathcal{H}$.

More formally, we have

**Definition D.3** (VC dimension). *Let $\mathcal{H}$ denote a class of functions on $X$ taking values in $\{-1, +1\}$. Define the growth function $\Pi_{\mathcal{H}} : \mathbb{N} \to \mathbb{N}$ as*

$$\Pi_{\mathcal{H}}(m) = \max_{\{x_1, \ldots, x_m\} \subset X} |\{(h(x_1), \ldots, h(x_m)) : h \in \mathcal{H}\}|.$$

*If $\Pi_{\mathcal{H}} = 2^m$ we say $\mathcal{H}$ shatters the set $\{x_1, \ldots, x_m\}$. The VC dimension of $\mathcal{H}$ is the size of the largest shattered set, i.e.*

$$VCdim(\mathcal{H}) = \max\{m : \Pi_{\mathcal{H}}(m) = 2^m\}$$

*If there is no largest $m$, we define $VCdim(\mathcal{H}) = \infty$.*

The VC dimension is appealing partly because it can be upper bounded for many classes of functions (see for example, [10]). Similar to the Rademacher complexity, the VC dimension is measured on the entire hypothesis space. The Geometric Complexity, in contrast, is instead measured for given

function within the hypothesis space allowing for more direct comparison between elements within the class $\mathcal{H}$. Computing the VC dimension for a given function set $\mathcal{H}$ may not be always convenient since, by definition, it requires computing the growth function $\Pi_{\mathcal{H}}(m)$ for all subsets of order $m \geq 1$; whereas, the Geometric Complexity relies only on first derivatives and is much easier to precisely compute.

### D.3 Sharpness and Hessian related measures

Another broad category of generalization measures concerns the concept of "sharpness" of the local minima; for example, see [44, 51, 68]. Such complexity measures aim to quantify the sensitivity of the loss to perturbations in model parameters. Here a flat minimizer is a point in parameter space where the loss varies only slightly in a relatively large neighborhood of the point. Conversely, the variation of the loss function is less controlled in a neighborhood around a sharp minimizer. The idea is that for sharp minimizers the training function is more sensitive to perturbations in the model parameters and thus negatively impacts the model's ability to generalize; see also [1] which argues that flat solutions have low information content.

The sharpness of a minimizer can be characterized by the magnitude of the eigenvalues of the Hessian of the loss function. However, since the Hessian requires two derivatives this can be computationally costly in most deep learning use cases. To overcome this drawback, [51] suggest a metric that explores the change in values of the loss function $f$ within small neighborhoods of points. More precisely, let $\mathcal{C}_{\epsilon}$ denote a box around an optimal point in the domain of $f$ and let $A \in \mathbb{R}^{n \times p}$ be a matrix whose columns are randomly generated. The constraint $\mathcal{C}_{\epsilon}$ is then defined as:

$$\mathcal{C}_{\epsilon} = \{z \in \mathbb{R}^p : -\epsilon(|(A^+ x)_i|) \leq z_i \leq \epsilon(|(A^+ x)_i| + 1) \quad \forall i \in \{1, 2, \ldots, p\}\}$$

where $A^+$ denotes the pseudo-inverse of $A$ and $\epsilon$ controls the size of the box. Keskar et al. [51] then define "sharpness" by

**Definition D.4** (Sharpness). *Given $x \in \mathbb{R}^n$, $\epsilon > 0$ and $A \in \mathbb{R}^{n \times p}$, the $(\mathcal{C}_{\epsilon})$-sharpness of $f$ at $x$ is defined as*

$$\phi_{x,f}(\epsilon, A) = \frac{(\max_{y \in \mathcal{C}_{\epsilon}} f(x + Ay)) - f(x)}{1 + f(x)} \times 100$$

Another related complexity measure is the *effective dimension* which is computed using the spectral decomposition of the loss Hessian [67]. Since the effective dimension relies on the Hessian, it causes flat regions in the loss surface to also be regions of low complexity w.r.t. this measure.

Effective dimensionality [66] was originally proposed to measure the dimensionality of the parameter space determined by the data and is computed using the eigenspectrum of the Hessian of the training loss.

**Definition D.5** (Effective dimensionality of a symmetric matrix). *The effective dimensionality of a symmeteric matrix $A \in \mathbb{R}^{k \times k}$ is defined as*

$$N_{\text{eff}}(A, z) = \sum_{i=1}^{k} \frac{\lambda_i}{\lambda_i + z}$$

*where $\lambda_i$ are the eigenvalues of $A$ and $z > 0$ is a regularization constant.*

When used in the context of measuring the effective dimension of a neural network $f(x; \theta)$ with inputs $x$ and parameters $\theta \in \mathbb{R}^k$, we take $A$ to be the Hessian of the loss function; i.e., the $k \times k$ matrix of second derivatives of the loss $\mathcal{L}$ over the data distribution $\mathcal{D}$ defined as $\text{Hess}_{\theta} = -\nabla^2 \mathcal{L}(\theta, \mathcal{D})$. Furthermore, the computation of the effective dimension involves both double derivatives (to compute the Hessian) but also evaluation of the eigenvalues of the resulting matrix. This can introduce a prohibitive cost in computation for many deep learning models.

The Geometric Complexity differs from these complexity measures in a meaningful way. Sharpness and effective dimension are ultimately concerned with the behavior and derivatives of the loss function with respect to the parameter space. The Geometric Complexity, however, is measured using derivatives of the learned model function with respect to the model inputs. Furthermore, the Geometric Complexity is computed using only a single derivative, making it computationally tractable to measure and track.

That being said, these sharpness measures and the Geometric Complexity are also quite related. For example, as explained by the Transfer Theorem in Section 5, for neural networks these flat regions are also the regions of low loss gradient and thus of low GC. Furthermore, similar to the GC, the effective dimension can also capture the double descent phenomena and, in [67], the authors argue the effective dimension provides an efficient mechanism for model selection.