# OpenReview forum: "Why neural networks find simple solutions:  The many regularizers of geometric complexity"
_NeurIPS.cc/2022/Conference — NeurIPS 2022 Accept_

### Official Review · Reviewer_UHLw · 2022-07-09

**Rating:** 6
**Confidence:** 3
**Soundness:** 3 good
**Presentation:** 3 good
**Contribution:** 3 good

**Summary:**

The authors introduce the Geometric Complexity (GC), a novel measure/predictor of neural network performance, based on the evaluation of a discrete Dirichlet energy on the logits in the output of the network. Given that the Dirichlet energy is a measure of how variable a function is, the Geometric Complexity is closely linked to many more common ML notions e.g. regularization and smoothness.
The authors argue that keeping a low GC, while training the NN model, allows a test performance improvement. This conclusion is drawn from the observation that a variety of well-known Deep Learning heuristics, including initialization, regularization and fine-tuning of the parameters of the learning algorithm seem to produce models with low GC that generalize well.
Finally, the authors show that GC can be used to trace the double-descent behavior, in the transition to the over-parametrized regime.

AFTER REBUTTAL
The authors have addressed most of the concerns raised in the review process and clarified and expanded upon their work.

**Questions:**

- Figure 3: The curve related to L2 norm does not seem to show the expected behavior, can the authors comment on it, or provide a plot where more values are shown in the range of regularization intensity where the GC drops?
- The accuracies shown in figure 4 seem to be way below the reference performance of a Resnet 18 on CIFAR10. The effect shown in the plot seems very clean, with lower GC and better generalization, but does this apply also close to state-of-the-art performance?
- Since GC can be explicitly evaluated and the authors argue that many heuristics implicitly minimize it, I think the authors should try a set of experiments where this minimization is done explicitly (even though the required computations may be very expensive). Otherwise, the correlation vs causation question remains open. Is there a reason why such a test was not performed, at least in a small network setting?
- Is there any other way of using the Dirichlet energy and harmonic theory to guide the learning process?
- Since the GC is data-dependent, how can it be used in the context of transfer learning and in OOD generalization settings? Could it still provide some useful insight, and would the smoothness vs generalization relationship still hold, even though the network was not trained on that data?
- Given two different DNN architectures, can their GC still be informative on which will perform best? Can GC be used for model selection?

**Limitations:**

I think that the main unaddressed limitation of this work is that the authors do not spend enough time trying to disprove their claim. This could only be achieved by explicitly controlling the GC and finding out whether its minimization is a sufficient drive for effective training.

**Strengths And Weaknesses:**

Computability is an essential property for any performance metric that is proposed to predict NN performance. The authors argue that the Dirichlet energy, while being conceptually close to other smoothness assessment metrics, is much simpler to estimate, as it is simply defined as an expectation over the training set. Moreover, the theoretical connection with harmonic theory allows for a very simple intuition of the concept of smooth interpolators and might be useful for designing better (more explicit) training algorithms.
What seems to be the main weakness of this work is that the assumption of the authors, of the fundamental importance of GC minimization in training DNNs, is only supported through a limited set of positive observations. The authors go through a range of well-known and functioning deep learning heuristics and show the relationship/correlation with a GC decrease. However, it remains unclear whether this decrease is a sufficient/necessary condition for achieving better performance. What would complete this study is a set of experiments where the GC is explicitly controlled and competitive performance is reached, either with an exact GC minimization (probably too computationally expensive), or with some novel regularization technique that aims at GC minimization.
Another weak point of this work is in the presented experiments, involving relatively small models and semi-obsolete datasets (MNIST, CIFAR).

---

> ### Author Response · Authors · 2022-08-02
> **Thanks for your review!**
>
> Thanks for your review. We are happy that you found “the theoretical connection with harmonic theory allows for a very simple intuition of the concept of smooth interpolators”  and that it “might be useful for designing better (more explicit) training algorithms.” This is also our belief! You’ll find below our best effort to answer your questions.
>
> We have taken steps to address your main concern about whether GC was beneficial in-and-by itself by performing explicit regularization experiments with GC (SM Section C.4) and repeated our Fig. 4 experiments in more competitive settings (SM Section C.6.) Please consider raising your score if you are satisfied with them.
>
> > *“Since GC can be explicitly evaluated and the authors argue that many heuristics implicitly minimize it, I think the authors should try a set of experiments where this minimization is done explicitly*
>
> We performed experiments adding GC regularization directly to the loss and show that decreased GC leads to increased test set performance (SM Section C.4). As now more clearly mentioned in the updated version (lines 215-219) explicit GC regularization is also known as Jacobian regularization in previous works that have shown the benefit of this type of regularization for increased test accuracy (See Tables III, IV, and V in Sokolic et al. 2017).
>
> >  *"Figure 3: The curve related to L2 norm does not seem to show the expected behavior, can the authors comment on it, or provide a plot where more values are shown in the range of regularization intensity where the GC drops?”*
>
> For the sake of brevity, we put all the explicit regularization on the same plot. The trade-off is we had to choose regularization rates for different regularization types in the same range. Now, this range does not fully coincide for all three types of regularization, which is particularly visible for L2. We have now added separate plots for L2 with a better choice of the regularization rates in SM Section C.5 (Fig. 20).
>
> > *“The accuracies shown in figure 4 seem to be way below the reference performance of a Resnet 18 on CIFAR10. The effect shown in the plot seems very clean, with lower GC and better generalization, but does this apply also close to state-of-the-art performance?”*
>
> To avoid masking effects, we studied training heuristics in isolation. That said, to address your concerns we conducted experiments with more of the bells of whistles found in competitive settings. We recover all the effects on GC discussed with SGD with momentum (SM Section C.7); almost all the effects with Adam, except for the batch size impact (SM Section C.8) -  we believe that changing the batch size affects the variability in the local loss surface seen by the optimizer at each step; since for Adam the local geometry of this surface determines the rescaling of the learning rate in every dimension, it affects also the pressure on GC in ways that are harder to predict. In SM Section C.6, we analyze the effect of the learning rate and batch size on GC in the presence of a learning rate schedule (which affects the pressure on GC during training) and data augmentation. In all instances, we still observe the general trend that hyper-parameter sweeps with higher test accuracy tend to come with lower GC.
>
> > *“Is there any other way of using the Dirichlet energy and harmonic theory to guide the learning process?”*
>
> We have observed that well-tuned neural networks naturally converge toward solutions that minimize the Dirichlet energy. These minimizers are known to be harmonic functions. The study of harmonic maps and harmonic map flow is already a well developed area of research and there are many fundamental properties and consequences that could be used to further shed light on the learning process when interpreted in this setting.
>
> > *“Since the GC is data-dependent, how can it be used in the context of transfer learning and in OOD generalization settings? ”*
>
> Looking at Figure 1, the neural network learns the function with minimal volume inside the data region. Outside of it, it is still very close to the parabola that generated the data very far away from the training set. This example suggests an interpolating function with minimal GC within the dataset that may guess a “probable” pattern outside of the data region that’s minimal in some sense.
>
> > *“Can GC be used for model selection?”*
>
> All of our experiments seem to point in this direction, i.e., that at the same level of train error the solutions with lower GC generalize better. Justifying this claim rigorously would require further investigation and we expect that it will depend heavily on the structure of the data distribution as well. From our preliminary investigation, we don’t believe it to be generally true for every data distribution, but mostly the ones that are generated by some form of “harmonic” process.

---

> > ### Comment · Reviewer_UHLw · 2022-08-06
> > **Response to Rebuttal**
> >
> > I would like to thank the authors for their answers and for their effort in trying to improve the manuscript.
> >
> > I still have some concerns that I'd like to present to the authors:
> > 1- I think the strong connection with previous works on Jacobian regularization should be pointed out earlier in the paper. In light of these works, can the GC really be presented as a novel metric?
> > 2- Too many of the results showing the positive correlation between GC and test score compare models that perform very poorly (compared to what can be achieved with a Resnet 18). This applies almost to all plots, except those that actually implement the known heuristics. In my opinion, this work can become relevant for ML practice only if GC turns out to be informative and discriminative also close to state-of-the-art performance, and if it can be used to improve scores or to devise more effective algorithms. Otherwise, showing that this correlation is strong when very bad models are compared can be a little misleading. In my opinion, the authors should focus on the region with 80+% accuracy.
> > 3- The experiments with direct GC regularization do not reach very good performance. Moreover, if I am not mistaken, both the GCs and the test scores are lower when compared to the models trained with all the known heuristics. This means that GC would not help select the best model among them, and this goes against the main message of this work. This seems like a crucial point to discuss.
> > 4- It would be interesting to have a plot that compares some of the optimal models found in each of the presented studies and see if in this comparison the GC is still telling with respect to the best generalization performance.
> > 3- Some of the plots in the supplementary seem incomplete.
> > 4- The discussion of the limitations of the work should definitely include that the authors do not believe that their claims are generally true for every data distribution. And they should expand on what they mean with data generated by some form of “harmonic” process.
> >
> > Thank you for your time!

---

> > > ### Author Response · Authors · 2022-08-08
> > > **Thanks for your response**
> > >
> > > In light of your new comments, we believe there is a significant misunderstanding:
> > >
> > > > "This means that GC would not help select the best model among them, and this goes against the main message of this work”
> > >
> > > With respect, that is not the main message of this work. The goal of our paper is to introduce a new complexity measure, GC, that is computationally tractable (Sec 2), captures the double-descent phenomena (Sec 6) and to show that this measure is, in fact, implicitly regularized through standard training procedures such as initialization (Sec 3), regularization (Sec 4), and common hyper-parameter tuning (Sec 5). We verify these claims through experiments and motivate them with theoretical arguments, uncovering a novel unifying simplicity bias in DL.
> > >
> > > Although our experiments suggest this simplicity bias improves test accuracy, and that GC may be used to select best models, we are not making the causal claim, nor it is the message of our work. The scope of such a claim would require its own investigation to understand when/how/if GC can be used to select models and would follow a different approach theoretically. There is no reference to “model selection” in our original submission. The term was only added to the future work section as a conjectural future direction, in response to your initial comments.
> > >
> > > > In light of these works [on Jacobian regularization], can the GC really be presented as a novel metric?
> > >
> > > Jacobian regularization is about a loss regularizer rather than a complexity measure. It is not our aim to devise regularizers. In fact, explicit GC regularization was not part of the original submission, but only added at your request.
> > >
> > > The novelty of our work resides in two main aspects: First, GC captures the double-descent phenomena; second, GC is implicitly regularized by a wide range of common training and tuning heuristics, which shows a novel simplicity bias in DL.
> > >
> > > >  this work can become relevant for ML practice only if GC turns out to be informative and discriminative also close to state-of-the-art performance, and if it can be used to improve scores or to devise more effective algorithms.
> > >
> > > Our paper is a theoretical paper aimed at establishing a unifying mechanism at play in existing and common training heuristics in DL. Although we believe that our findings will be useful in future works to improve SOTA, training procedures, and model selection, it is not our goal here. Our goal is to understand the mechanism.
> > >
> > > > the authors should focus on the region with 80+% accuracy.
> > >
> > > To  scientifically observe the effect for a given training heuristic on GC, we study it in isolation to remove possible sources of confounding effects as much as possible.
> > >
> > > To achieve 80+% accuracy on CIFAR10, several heuristics need to be combined making it harder to disentangle which heuristic is responsible for the effect.
> > >
> > > This is particularly visible in the learning curves in C.6, Fig. 23 where we achieve 80+% as requested when using multiple heuristics - we note that the observed correlation low GC / high test accuracy persists
> > >
> > > > The experiments with direct GC regularization do not reach very good performance. Moreover, both the GCs and the test scores are lower when compared to the models trained with all the known heuristics
> > >
> > > Again, our goal is not to devise novel or performant regularizers
> > >
> > > That said,  we disagree with the statement.
> > >
> > > Our MNIST experiment in Fig.18 with explicit GC regularization achieved highest test accuracy (0.9857) and lowest GC (0.025679) than any of the MNIST experiments (Sec C.2.). Additionally, Fig.19 shows that GC as an explicit regularizer is beneficial also with CIFAR10. However, it is not reflective of the highest test accuracy that could be achieved: each model had to be stopped before peak test accuracy because of time limitation.
> > >
> > > > Some of the plots in the supplementary seem incomplete.
> > >
> > > Can the reviewer kindly clarify which plots and how exactly they are incomplete so we can properly address them.
> > >
> > >
> > > > The discussion of the limitations of the work should definitely include that the authors do not believe that their claims are generally true for every data distribution.
> > >
> > > We disagree with this statement as there must be some misunderstanding.
> > >
> > > None of the claims we make in this paper are data-distribution dependent. The only assumption we make is a very general assumption on the form of the loss that covers both least-square and cross-entropy losses, which we already state as limitation.
> > >
> > > The reviewer seems to confuse the conjectural response to the reviewer's conjectural question (“Can GC be used for model selection?”) with our actual paper claims. Again, answering this question is not the goal of the current paper. This is future work and we believe that a full answer to that future question is likely to involve consideration about the data distribution.

---

> > > > ### Comment · Reviewer_UHLw · 2022-08-08
> > > > **Reply**
> > > >
> > > > I sincerely thank the authors for this response. Likely because of a personal research bias, my interpretation of the results and of the message of the paper was indeed off. I was trying to force them into something different than the actual content of the paper.
> > > >
> > > > Although I still find the proposed (and discussed) direction interesting I now understand that the connection between GC complexity and generalization performance is not meant to be completely understood and explained in this work, and the potential open questions and inconsistencies will need to be addressed in future research.
> > > > The evidence for the GC minimization being a common underlying trait of many well known learning heuristics is indeed convincing, and I see that this is a sufficiently interesting point in itself.
> > > >
> > > > Still, I hope that at least part of this discussion was not completely pointless, and that mentioning some possible directions of development of this work will increase its impact.
> > > >
> > > > I plan to increase my score to a positive evaluation in view of these clarifications, which solved my misunderstanding. Thanks again to the authors for their efforts in explaining and clarifying their work.

---

> > > > > ### Author Response · Authors · 2022-08-09
> > > > > **Thank you!**
> > > > >
> > > > > We'd like to thank you for your thorough response. We are very glad that we were able to clear the misunderstanding and that you find the current work interesting. The question of fully understanding how GC affects generalization is certainly an interesting one as well and something to focus on for future work. This discussion has certainly been very helpful and we appreciate your thoughtful reviews and comments. Thank you for offering to raise your score.

---

### Official Review · Reviewer_jmMn · 2022-07-11

**Rating:** 7
**Confidence:** 4
**Soundness:** 3 good
**Presentation:** 3 good
**Contribution:** 4 excellent

**Summary:**

Regularization is key when it comes to model selection; if we seek models that generalize well, choosing the less complex one tends to be a good heuristic in classical machine learning. In deep learning, however, it is not clear what is a good metric to measure the "complexity" of a given model, as the classic ones tend to be unreliable (e.g. parameter counting and the double-descent effect), and/or intractable (e.g. VC dimension).

This work presents a new complexity metric, the Geometric Complexity (GC). Throughout the paper, the authors present their metric in simple scenarios (ReLU and linear networks), relate it with other metrics (Lipschitz smoothness and Dirichlet energy), and empirically demonstrate that many common regularization techniques in DL minimize the GC under the hood. Finally, the authors reproduce the original double descent experiments, and show that GC recovers the traditional U-shaped curve.

**Questions:**

**Questions (Q)**
1. What is the training error in section 6? I assume that it is zero, but otherwise I think it is important to show the training error as well. Given that the U-shaped curve explains overfitting, I don't find useful to recover the curve with GC if that does not explain overfitting as in traditional ML.

**Nitpicks (N)**
1. The quantity $h/B$ in line 246 has been studied before, maybe you want to add a few lines commenting on it. See, e.g., [Accurate, Large Minibatch SGD: Training ImageNet in 1 Hour](https://arxiv.org/pdf/1706.02677.pdf), [Don't Decay the Learning Rate, Increase the Batch Size](https://arxiv.org/pdf/1711.00489.pdf), or [An Empirical Model of Large-Batch Training](https://arxiv.org/pdf/1812.06162.pdf).
2. The font of the figures should be increased.
3. You should point to specific parts of the SM in the main text, not just the SM in general.


**Limitations:**

I think the authors correctly addressed the limitations of the method.

**Strengths And Weaknesses:**

**Strengths (S)**
1. The main contribution of the paper can be fairly impactful. GC is computable, and it seems to be well correlated with the test error. More importantly, previous work does not need to be thrown away. The authors related GC with other metrics, and showed that effectiveness regularization techniques greatly affect the GC. I expect this work to be impactful.
2. The paper has a strong empirical flavour, which I find necessary for a paper on this topic. Moreover, the range of methods tested is broad and is well-structured in different sections (initialization, implicit regularization, and explicit regularization).
3. The theoretical results and claims look sound to me. Specially, the connection of GC with Dirichlet energy is really appealing. However, I have only fully read the main paper and the proofs until A3.
4. The paper is well written and easy to follow.

**Weaknesses (W)**
1. While I agree with the authors on removing extra sources of noise for the main experiments (e.g. not using momentum or lr schedulers), there should a section/experiment where all the DL machinery is put together, and compare the GC of those new models with the ones already on the paper. Otherwise, the performance gap is too big compared to other works. For example, test accuracy on CIFAR10 is around 75% in the paper (Figure 4), while using the same model **other works achieve an accuracy of 95.55%** (according to [papers with code](https://paperswithcode.com/sota/image-classification-on-cifar-10?tag_filter=3)).
2. I feel the paper glosses over connections with the proposed metric. For example, line 186 says that Jacobian regularization basically _minimizes the proposed metric (GC) explicitly_ on the loss, but the topic is covered with 4 lines. This is _really_ important, Jacobian regularization is not just another regularization method to mention. Similarly, section 7 says that GC is similar to Kolmogorov complexity and MDL, but these similarities are not explained anywhere.
3. I miss some motivation and intuition for the proposed metric, Def. 2.1 feels arbitrary until you read the end of section 2. I think introducing the metric as a modification of Dirichlet energy minimization would feel more natural.

_Note:_ While I can understand and follow all the paper, I am not that well-versed in the topic, and therefore I cannot fully assess the novelty of the proposed metric as I could miss well-known literature. To me, the metric seems novel for this purpose, but strong connections were vaguely mentioned, which I find concerning.

---

> ### Author Response · Authors · 2022-08-02
> **Thanks for your review!**
>
> Thanks for your review. We are very happy that you “expect this work to be impactful” and you found “the range of methods tested broad and  well-structured” and that “the theoretical results and claims look sound” to you. We are glad that “the connection of GC with Dirichlet energy is really appealing” to you (it is to us too!!!). You’ll find below our best effort to answer your questions. Please consider raising your score if you are satisfied with them.
>
> > *“While I agree with the authors on removing extra sources of noise for the main experiments (e.g. not using momentum or lr schedulers), there should be a section/experiment where all the DL machinery is put together”*
>
> To address your requests we repeated our experiments on more competitive settings and our conclusions hold: 1) we replicated our experiments using optimizers that are more commonly used in SOTA settings (SM Section C.7, and Section C.8), and 2) we conducted additional experiments using more of the bells and whistles commonly used in SOTA settings, such as data augmentation and learning rate schedules (SM Section C.6), which gave us a 10% boost to our previous test accuracy (to reach performance SOTA levels we’d need more extensive hyper-parameter sweeps than time allows).
>
> For the first direction, when considering SGD with momentum (SM Section C.7), all our conclusions hold as before. We see similar outcomes when using Adam (SM Section C.8) though the influence of batch size on GC is more complex. (We believe that changing the batch size affects the variability in the local loss surface seen by the optimizer at each step. Since for Adam the local geometry of this surface determines the rescaling of the learning rate in every dimension, it affects also the pressure on GC in ways that are harder to predict.) For both sets of experiments, we observe that trend where higher test accuracy tends to come with lower GC.
>
> For the second direction, we still observe the trend that hyper-parameter sweeps achieving higher test accuracy also tend to achieve lower GC (SM Section C.6); however, the learning curves are harder to interpret than those in Fig. 4. because of the many mechanisms interacting in complex ways with each other. Similarly the effect of batch size and learning rate on lowering GC is visible.
>
> > *“I feel the paper glosses over connections with the proposed metric. For example, line 186 says that Jacobian regularization basically minimizes the proposed metric (GC) explicitly on the loss, but the topic is covered with 4 lines. [...] Similarly, section 7 says that GC is similar to Kolmogorov complexity and MDL, but these similarities are not explained anywhere.”*
>
> Concerning the relation between GC and Jacobian regularization, we rewrote and expanded this section in the updated version (lines 211-223). We also conducted additional experiments where the GC is directly regularized, recovering the results from previous works on Jacobian regularization (SM Section C.4).
>
> Concerning the relation between GC and Kolgomorov complexity or minimal length, we rewrote the corresponding paragraph (lines 297-311) describing our intuition about the relationship between these complexity measures. We clarified that this relationship is conjectural and a topic for future work.
>
> > *“To me, the metric seems novel for this purpose, but strong connections were vaguely mentioned, which I find concerning.”*
>
> We clarified the connections with other complexity which we believe are related (lines 297-311).  Upon acceptance, we will add a section to the supplementary material with the definitions of the standard and more recent complexity measures and how they relate to and differ from GC more explicitly.
>
> > *“What is the training error in section 6? I assume that it is zero, but otherwise I think it is important to show the training error as well. Given that the U-shaped curve explains overfitting, I don't find useful to recover the curve with GC if that does not explain overfitting as in traditional ML.”*
>
> All the models were fully trained. To address your comments, we've added the training loss curves to the plots in Figure 5. We indeed recover the traditional U-curve test/train loss overfitting pattern if we replace the network width with the geometric complexity on the x-axis (Figure 5, right).
>
> > *“The quantity h/B in line 246 has been studied before, maybe you want to add a few lines commenting on it. See, e.g., Accurate, Large Minibatch SGD: Training ImageNet in 1 Hour, Don't Decay the Learning Rate, Increase the Batch Size, or An Empirical Model of Large-Batch Training.”*
>
> Thanks! We added the references (lines 263-264).
>
> > *"The font of the figures should be increased."*
>
> Thanks! We'll do upon acceptance for the camera ready when more space is available.
>
> > *"You should point to specific parts of the SM in the main text, not just the SM in general."*
>
> Thanks! We updated the references to the SM with precise mention of the sections.

---

> > ### Comment · Reviewer_jmMn · 2022-08-03
> > **Thanks for the changes**
> >
> > Thank you for the nice reply and the accompanying changes to the manuscript. I am really pleased with the new changes, and I think the paper has significantly improved (specially the double descent section, now it is way clearer).
> >
> > Regarding the score, I anticipated (based on the quality of the paper) that the authors were going to respond accordingly to the changes requested, so I am happy to see that I was right, yet I believe the score is appropriate at the moment. Having said that, I might consider changing my score after discussing with the rest of the reviewers.
> >
> > Thanks once again and congrats for the great response!

---

### Official Review · Reviewer_wsiC · 2022-07-11

**Rating:** 7
**Confidence:** 2
**Soundness:** 3 good
**Presentation:** 3 good
**Contribution:** 3 good

**Summary:**

Summary

The paper proposes a new notion of complexity measure called geometric complexity (GC) . The paper discusses how GC relates to the familiar concepts in deep learning literature and made extensive links to existing regularizer methods. I enjoyed a lot reading this paper, and found it helpful to understand the connections among all the mentioned regularizers.




**Questions:**

Please see the above comments for questions.

**Limitations:**

The author has addressed the limitations.

**Strengths And Weaknesses:**

Major strength
1. The paper is well organized and well written.
2. The authors made extensive analysis to link GC with typical deep learning architectures, demonstrating the connections between existing training regularizers and GC. These links explicitly demonstrate that GC , to some extent,  unify the regularizers in the sense that penalizing these regularizers would simultaneously penalize GC.
3. In comparison to other complexity measures, GC is more computationally tractable.
The paper empirically evaluates the connection between the generalization property and GC: after the critical region, increasing model size would necessarily decrease GC, showing the usefulness of GC in estimating the generalization error, showing potential benefits of using GC to improve generalization.

Questions.
I am not expert on this complexity measurement topic, but I have some basic understanding of complexity measures. If you don’t mind me asking, what is the advantage of GC over Rademacher complexity? Is it because GC is more computationally convenient?  Is it possible to compare these two types of complexity in some sense empirically or theoretically?

---

> ### Author Response · Authors · 2022-08-02
> **Thanks for your review!**
>
> Thanks for your review. We are very pleased that you “enjoyed a lot reading this paper, and found it helpful to understand the connections among all the mentioned regularizers” and “ well organized and well written.” You’ll find below our best effort to answer your questions. Please consider raising your score if you are satisfied with them.
>
> > *“What is the advantage of GC over Rademacher complexity? Is it because GC is more computationally convenient? Is it possible to compare these two types of complexity in some sense empirically or theoretically?”*
>
>
> Complexity measures can be broadly classified into two classes as whether they measure the complexity of the whole hypothesis space or whether they measure the complexity of individual functions. The Radamacher complexity measures the complexity of the whole hypothesis space, while GC measures the complexity of individual functions in that space. Measuring the complexity of the whole space is challenging in deep learning for two reasons:
> 1) theoretically, because the hypothesis space for a large neural network is extraordinarily complex (measures like the Radamacher complexity measure the space complexity by looking at some notion of the “most complex” hypothesis in the space, which are in general not the ones “selected” by the training procedure in deep learning),
> 2) practically, complexity measures defined on the whole space are hard to compute when the space is big, especially when they involve taking a supremum on all the functions in the space as for the Radamacher complexity. An advantage of complexity measures like GC is that we can compute them at every step of the learning process and understand by looking at the learning curves whether a given training procedure favors simple solutions more so than complex ones. Complexity measures focusing on the whole space like the Radamacher complexity have a harder time assessing complexity biases for training procedures in terms of the simplicity of the outcomes.
>
> We briefly touched upon the difference between GC  and Radamacher complexity in the introductory paragraph in Section 2, but space considerations didn’t allow us to develop more and make the point very clear. We added a sentence clarifying the difference between GC and the Radamacher complexity in lines 77-79 in the updated version. We will also add to the appendix a new section with the precise definitions of the main complexity measures and how they compare to GC upon acceptance.

---

> > ### Comment · Reviewer_wsiC · 2022-08-09
> > **Thanks for your response**
> >
> > Thanks for the nice response from the authors! Given the fact the authors have addressed my questions, I will maintain my score.

---

### Author Response · Authors · 2022-08-02
**Thanks for your reviews!**

Thank you for your kind and balanced reviews. We are grateful that you “enjoyed a lot reading this paper”, found it “helpful to understand the connections among all the mentioned regularizers”, and that you “expect this work to be impactful.” It is nice to hear that you found “the connection of GC with Dirichlet energy really appealing” and that “the theoretical connection with harmonic theory allows for a very simple intuition of the concept of smooth interpolators”. It’s also our belief that these connections “might be useful for designing better (more explicit) training algorithms.”

We want to underline that GC is a unique complexity measure in that: it can be easily computed for neural networks (see Section 2); it has been connected with generalization extensively in the submission (See Section 4, Section 5, and Section 6) and in the additional experiments we provide in the rebuttal (SM Section C.4 to C.8); it has been connected with many aspects of the NN ecosystem: initialization (Section 3), optimization (Section 5, and SM Section C.7 and C.8), other regularization methods (Section 4) .

We tried our very best below to satisfy your requests and answer your questions within the time frame imparted. We think that as a result the uploaded revised paper is now clearer and stronger thanks to your comments. (We highlighted the new parts in red for your convenience.)

We conducted additional experiments and put their results in the Supplementary Material (Sections C.4, C.5, C.6, C.7, and C.8)  to support our responses. We show that:
* Our results remain valid for optimizers like SGD with momentum and Adam (SM Section C.7 and C.8), learning rate scheduling, and data augmentation (SM Section C.6), all techniques used in competitive  settings
* Direct explicit regularization with GC is beneficial to increase the test accuracy by itself (SM Section C.4).
* With parameter sweeps targeted for individual explicit regularization, including L2 (SM Section C.5) the curves are clearer.

Please consider raising your scores in consequence if your requests have been met and questions answered.

---

### Meta-Review · Area_Chair_tG9T · 2022-08-23

**Recommendation:** Accept
**Confidence:** Certain

**Metareview:**

All reviewers and the AC find this paper makes valuable contributions to the deep learning theory community. Thus, the AC recommends acceptance.

**Award:**

No

---

### Decision · Program_Chairs · 2022-09-14

Accept